# LEARNING TO SOLVE BILEVEL PROGRAMS WITH BINARY TENDER

**Bo Zhou, Ruiwei Jiang, Siqian Shen**
Department of Industrial and Operations Engineering
University of Michigan, Ann Arbor, MI 48109, USA
`{bozum, ruiwei, siqian}@umich.edu`

## ABSTRACT

Bilevel programs (BPs) find a wide range of applications in fields such as energy, transportation, and machine learning. As compared to BPs with continuous (linear/convex) optimization problems in both levels, the BPs with discrete decision variables have received much less attention, largely due to the ensuing computational intractability and the incapability of gradient-based algorithms for handling discrete optimization formulations. In this paper, we develop deep learning techniques to address this challenge. Specifically, we consider a BP with binary tender, wherein the upper and lower levels are linked via binary variables. We train a neural network to approximate the optimal value of the lower-level problem, as a function of the binary tender. Then, we obtain a single-level reformulation of the BP through a mixed-integer representation of the value function. Furthermore, we conduct a comparative analysis between two types of neural networks: general neural networks and the novel input supermodular neural networks, studying their representational capacities. To solve high-dimensional BPs, we introduce an enhanced sampling method to generate higher-quality samples and implement an iterative process to refine solutions. We demonstrate the performance of these approaches through extensive numerical experiments, whose lower-level problems are linear and mixed-integer programs, respectively.

## 1 INTRODUCTION

Bilevel programs (BPs) are appealing for modeling problems that involve sequential decision interactions from two or multiple players. Their hierarchical decision processes arise in a wide range of real-world applications, including energy (Zhou et al., 2023), security (Bhuiyan et al., 2021), transportation (Santos et al., 2021), and market design (Nasiri et al., 2020). Recently, BPs have also shown strong modeling power in machine learning problems, including meta learning (Wang et al., 2021), actor-critic reinforcement learning (Hong et al., 2020), hyperparameter optimization (Bao et al., 2021), and deep learning (Chen et al., 2022). Next, we provide a brief introduction to BP, its generic formulation, and solution methods.

**Bilevel Program.** In a BP, a leader and a follower solve their own decision making problems in an interactive way: The leader's decisions made in the upper level will affect the follower's problem solved at the lower level (e.g., the leader's decisions are involved in the objective function and/or constraints of the follower's problem) and vice versa. This can be described formally as

$$\min_x f(x, y^*) \tag{1a}$$

$$\text{s.t. } x \in X(y^*) \tag{1b}$$

$$\text{where } y^* \in \arg\max g(y, x) \tag{1c}$$

$$\text{s.t. } y \in Y(x), \tag{1d}$$

where $x$ represents the leader's decision and $y$ represents the follower's. Here, the leader's decision $x$ will affect the follower's objective function $g(y, x)$ and feasible region $Y(x) \subseteq \mathbb{R}^m$. On the other hand, the leader's objective function $f(x, y^*)$ depends on both her own decision $x$ and the follower's *optimal* decision $y^*$, which is a function of $x$ defined through (1c)–(1d). Additionally, the leader's feasible region $X(y^*) \subseteq \mathbb{R}^n$ can also depend on $y^*$.

**Single-Level Reformulation.** To solve the BP (1), a general approach is to incorporate the lower-level decisions and their feasible region into the upper-level problem, giving rise to

$$\min_x f(x, y) \tag{2a}$$

$$\text{s.t. } x \in X(y) \tag{2b}$$

$$y \in Y(x). \tag{2c}$$

Formulation (2) relaxes the optimality of $y$, and therefore, its solution only provides a lower bound for the BP (1). To retrieve optimality, one can incorporate an optimality condition

$$g(y, x) \geq \phi(x), \tag{3}$$

where $\phi(x) := \max_{y \in Y(x)} g(y, x)$ represents the optimal value of the lower-level problem as a function of $x$. Hence, combining (2) and (3) produces a single-level reformulation of the BP (1). In case the lower-level problem is continuous (i.e., $y$ consists of continuous decision variables only), constraint (3) can be made explicit using strong duality or the KKT conditions. However, such luxury is immediately lost if the lower-level problem involves discrete decision variables (e.g., $y$ is binary or mixed-binary). In that case, the closed-form expression of $\phi(x)$ is either non-existent or highly intractable, prohibiting solving the BP effectively.

In light of this challenge, we study BPs with binary tender. Here, "tender" is defined as the linking variables between the upper- and lower-level problems and "binary tender" means that all tender variables are binary. Note that the problems in both levels can involve general decisions (e.g., continuous and/or integer variables), and we only assume that the entries of $x$ appearing in the lower-level formulation are binary-valued. Such BPs arise in many applications, including energy system expansion planning (Kabirifar et al., 2022), charging station planning (Li et al., 2022), competitive facility location (Qi et al., 2022), and network interdiction (Smith & Song, 2020). In addition, we assume the leader's feasible region is independent of $y^*$, i.e., $X(y^*) \equiv X$, which is a special case of (1). In this case, we can obtain an upper bound for the BP (1) from any feasible solution $x$ and the corresponding follower's optimal solution $y^*$ (e.g., by solving the follower's problem using $x$). Our main contributions are three-fold.

- We employ neural networks to learn and approximate the value function $\phi(x)$. Then, we derive a closed-form representation of the learned value function. This yields a single-level, mixed-integer formulation for the BP (1), which can be readily solved by off-the-shelf optimization solvers.

- Motivated by the fact that $\phi(x)$ is supermodular for a large class of BPs, we design an input supermodular neural network (ISNN) that ensures a supermodular mapping from input to output. We analyze the representability of both general neural networks (GNN) and ISNN to provide guidance for network architecture selection.

- To solve high-dimensional BPs, we propose an enhanced sampling method for generating higher-quality samples and training neural networks to better approximate $\phi(x)$ around the optimal solution. Building upon this enhanced sampling method, we execute an iteration process to improve the accuracy of the derived solution.

The remainder of the paper is organized as follows. In Section 2, we review related works in the literature. In Section 3, we elaborate our methodology of learning to solve BPs. We conduct numerical experiments in Section 4 and draw conclusions in Section 5.

## 2 RELATED WORKS

The theories and algorithms for finding optimal solutions to BPs depend on the structure and properties of the lower-level problem, as well as the coupling between the two levels (Kleinert et al., 2021; Beck et al., 2022).

**Bilevel linear/convex programs.** The earlier studies on BPs focused on linear or convex lower-level problems. Consequently, one can use the KKT conditions of the lower-level problem or strong duality to reformulate the BP as a single-level problem with complementarity constraints (Fortuny-Amat & McCarl, 1981) or bilinear terms (Zare et al., 2019; McCormick, 1976), respectively. In both cases, solving BPs boils down to solving the resulting single-level, nonconvex, and nonlinear reformulation (Colson et al., 2005; Bard, 1998).

**Bilevel (mixed-integer) nonconvex programs.** Recently, more studies focused on more general BPs having discrete decision variables or nonconvex objectives/constraints. In such problems, we notice that, because of the existence of integer decision variables or nonconvexity, neither KKT conditions nor strong duality approaches may be able to capture the (parametric) global optimal solutions at the lower level. To achieve global optimum, we instead need to exploit special structures of BPs (Qi et al., 2022) or construct the aforementioned relaxation (2) for single-level reformulation. For BPs with nonconvex objectives/constraints, approximation methods were employed to approximate the lower-level problem and reformulate the BP as a single-level formulation, such as polyhedral approximation (Sato et al., 2021) and other gradient-based methods in machine learning (Hong et al., 2020; Chen et al., 2022). However, when discrete decision variables are incorporated, gradient-based methods are no longer applicable. To address this issue, integer programming techniques are employed and various cutting plane-based algorithms, such as the MiBS solver, have been developed (Ralphs et al., 2015; Zeng & An, 2014). However, learning techniques have not been explored in solving such BPs, which is the focus of this paper.

**Embedding neural networks into optimization.** Due to its strong power in data analysis and fitting, machine learning, such as neural networks, has drawn wide interest in function approximation (Fujimoto et al., 2018; Liang & Srikant, 2017; Ferrari & Stengel, 2005). Furthermore, when the adopted activation function is a piecewise linear function, such as ReLU, the output $\tilde{\phi}(x)$ of a neural network can be reformulated as a mixed-integer linear form (Fischetti & Jo, 2018; Serra et al., 2018). Consequently, in case $\phi(x)$ is unknown or hard to access, one can use $\tilde{\phi}(x)$ to replace $\phi(x)$ and directly incorporate the mixed-integer representation of $\tilde{\phi}(x)$ into optimization formulations. Molan & Schmidt (2022) considered this idea in BPs with unknown lower-level problems. They utilized a neural network to approximate the mapping from $x$ to the optimal $y^*$ in the lower level, and exploited Lipschitz optimization techniques to reformulate the activation function (ReLU). Different from Molan & Schmidt (2022), we propose to learn and approximate $\phi(x)$ in BPs.

**Input convex neural network (ICNN).** Another related stream of literature proposed ICNNs, which ensure that the output approximation $\tilde{\phi}(x)$ is a convex function. In this case, the target optimization formulation remains a convex program if it was so before incorporating $\tilde{\phi}(x)$. ICNN was first proposed by Amos et al. (2017), who derived sufficient conditions on the neural network architecture and parameter settings for the convexity of $\tilde{\phi}(x)$. ICNNs were then applied to, e.g., optimal control (Chen et al., 2019) and energy optimization (Bünning et al., 2021). Different from these works, we study BPs with binary tender, and thus the input $x$ of the neural network is binary-valued. Accordingly, we consider supermodularity, the counterpart of convexity in discrete optimization, and establish ISNN that guarantees to output a supermodular $\tilde{\phi}(x)$. In Section 3, we derive a neural network architecture for ISNN and show its representability.

## 3 METHODOLOGY

We use neural networks (see Fajemisin et al. (2023)) to obtain an approximate value function $\tilde{\phi}(x)$ of $\phi(x)$ and solve the BP (1) by incorporating a closed-form expression of $\tilde{\phi}(x)$ into formulation (2)–(3). Figure 1 illustrates the proposed method, including sampling, training, and solving.

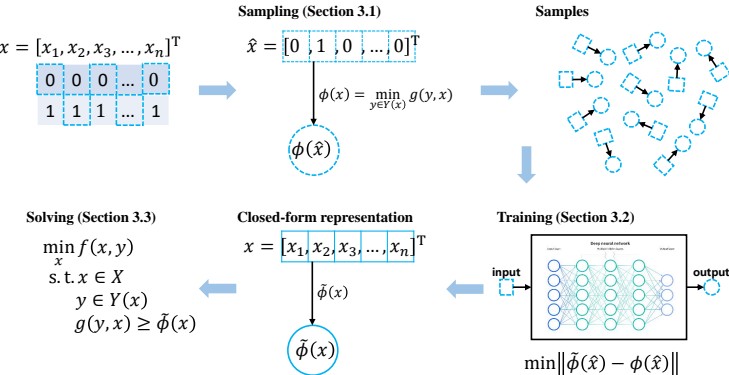

Figure 1: The overview of the proposed method.

## 3.1 SAMPLING

We consider a set of the upper-level decision variables $\hat{x}$ and compute $\phi(\hat{x}) = \max_y \{g(y, \hat{x}) : y \in Y(\hat{x})\}$ to obtain sample-label pairs $(\hat{x}, \phi(\hat{x}))$. When the dimension $n$ of $x$ is small, we can enumerate all possible sample-label pairs. As $n$ increases, the number of all sample-label pairs increases exponentially and enumeration becomes numerically prohibitive. In this case, we can sample a sufficiently large set of $\hat{x}$ instead. However, a naïve sampling may frequently produce infeasible solutions, let alone obtaining high-quality ones. To this end, we propose an enhanced sampling method to quickly find feasible samples and improve the quality of samples when the number of samples is limited.

**Enhanced Sampling.** A basic idea in enhanced sampling is "sampling via optimization." This involves confining the sample selection within the feasible region of the BP with respect to a random but tractable objective function. This is mathematically specified as

$$\min \ x^{\mathrm{T}}Qx + h^{\mathrm{T}}x \tag{4a}$$

$$\text{s.t. } x \in X, y \in Y(x), \tag{4b}$$

where $Q$ and $h$ are randomly generated $n \times n$ positive semi-definite matrix and $n \times 1$ vector, respectively. Accordingly, we can efficiently generate samples by solving (4). We note that the reason for solving a quadratic program here is to avoid repeated samples and enhance sampling efficiency.

Another basic idea is to sample more often in the vicinity of the optimal solution to the BP. A more accurate approximation $\tilde{\phi}(x)$ to the true $\phi(x)$ in this vicinity, even at the cost of inaccuracy for those $x$ far from optimum, enhances the tightness of approximating the BP (1), because most of the effort in solving (1) is spent on comparing the near-optimal $x$'s. To this end, we let $f_{ub}$ represent a known upper bound of the BP (1), which can be obtained from any feasible solution. Then, we strengthen (4) to be

$$\min \ x^{\mathrm{T}}Qx + h^{\mathrm{T}}x \tag{5a}$$

$$\text{s.t. } f(x, y) \leq f_{ub} \tag{5b}$$

$$x \in X, y \in Y(x), \tag{5c}$$

where constraint (5b) restricts the search space to elevate quality of samples. Here we note that though (5) is a mixed-integer quadratic program, we only need a feasible solution to (5) for sampling, which does not incur too much computational burden and saves the overall computational time. Furthermore, by sampling feasible solutions $\hat{x}$ to the BP, we can compute their objective value $f(\hat{x}, \hat{y}^*)$, $\hat{y}^*$ representing the corresponding optimal solution from the lower level, and iteratively strengthen $f_{ub}$ if $f(\hat{x}, \hat{y}^*) < f_{ub}$. This allows to continuously refine the search space and obtain higher-quality samples. Algorithm 1 summarizes the above sampling process.

---

**Algorithm 1** Enhanced Sampling

---

1: **INPUT**: sample size $N_s$, initial upper bound $f_{ub}$, maximum number of updates $N_{ub}$.
2: Initialize the set of samples $\Omega \leftarrow \varnothing$, count of samples $k_s \leftarrow 0$, count of updates $k_{ub} \leftarrow 0$.
3: **repeat**
4:     Randomly generate an $n \times n$ positive semi-definite matrix $Q$ and an $n \times 1$ vector $h$.
5:     Solve (5a)–(5c) and store an optimal solution $\hat{x}$.
6:     **if** $\hat{x} \notin \Omega$ **then**
7:         Solve $\max_{y \in Y(\hat{x})} g(y, \hat{x})$, and store the optimal value $\phi(\hat{x})$ and an optimal solution $\hat{y}^*$.
8:         Augment $\Omega \leftarrow \Omega \cup \{\hat{x}\}$ and $k_s \leftarrow k_s + 1$.
9:     **end if**
10:     **if** $k_{ub} < N_{ub}$ and $f(\hat{x}, \hat{y}^*) < f_{ub}$ **then**
11:         Update $f_{ub} \leftarrow f(\hat{x}, \hat{y}^*)$.
12:         $k_{ub} \leftarrow k_{ub} + 1$.
13:     **end if**
14: **until** $k_s = N_s$
15: **OUTPUT**: sample-label pairs $(x, \phi(x))$ for all $x \in \Omega$.

---

## 3.2 TRAINING

Using the sample-label pairs from Algorithm 1, we adopt supervised learning to train a neural network to fit the mapping $\phi(x)$. In this step, we consider two types of neural networks, i.e., GNN and

ISNN, which use the same architecture as shown in Figure 2. There are $K$ hidden layers and one output layer in the architecture and we employ passthrough to enhance the representability of the neural network. We note that the input of the architecture, $\tilde{x}$, can be different from $x$ (see Sections 3.2.1–3.2.2 for details). In addition, $\tilde{\phi}$ denotes the output of the neural network, $z_k$ denotes the output of the $k$th hidden layer, $W_k/D_k$ and $b_k$ are the weights and biases of the $k$th layer, respectively, and $\sigma(\cdot)$ denotes the activation function.

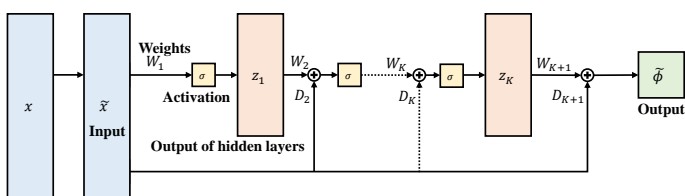

Figure 2: Neural network architecture.

Specifically, the neural network defines $\tilde{\phi}(\tilde{x})$ through

$$z_1 = \sigma(W_1 \tilde{x} + b_1) \tag{6a}$$

$$z_k = \sigma(W_k z_{k-1} + b_k + D_k \tilde{x}), \quad \forall k = 2, \ldots, K, \tag{6b}$$

$$\tilde{\phi} = W_{K+1} z_K + b_{K+1} + D_{K+1} \tilde{x}. \tag{6c}$$

### 3.2.1 GENERAL NEURAL NETWORK

For a GNN, we define $\tilde{x} := x$. Proposition 1 shows the representability of the GNN.

**Proposition 1.** *Consider an arbitrary $\phi : \{0, 1\}^n \to \mathbb{R}$. Then, the following statements hold.*
i. **(Universal Approximation)** *there exists a GNN with the architecture in Figure 2 such that $\tilde{\phi}$ fits $\phi$ exactly, i.e., $\tilde{\phi}(x) = \phi(x)$ for all $x \in \{0, 1\}^n$.*
ii. **(Maximum Representability)** *Suppose that the hidden layers of the GNN consist of $N_{nr}$ many neurons, which may be distributed arbitrarily among these layers. If we measure the GNN representability by the number of parameters it trains, then the representability is maximized when GNN consists of two hidden layers, i.e., when $K = 2$. In particular, the first layer consists of $N_{nr}/2$ (if $N_{nr}$ is even) or $(N_{nr} - 1)/2$ neurons (if $N_{nr}$ is odd), and the second layer consists of $N_{nr}/2$ (if $N_{nr}$ is even) or $(N_{nr} + 1)/2$ neurons (if $N_{nr}$ is odd).*
iii. **(Sufficient Fitting)** *Given $N_s$ sample-label pairs, the GNN is able to output an approximation $\tilde{\phi}$ that fits these pairs exactly if*

$$N_{nr} \geq \left\lceil \sqrt{(2n+2)^2 + 4N_s + 1} - (2n+3) \right\rceil. \tag{7}$$

We give a proof of Proposition 1 in Appendix A.1. According to Proposition 1, when we adopt a GNN for training, we incorporate two hidden layers in the GNN architecture, each with $N_{nr}/2$ neurons, where $N_{nr}$ is the smallest even number that satisfies (7).

### 3.2.2 INPUT SUPERMODULAR NEURAL NETWORK

In light of the special class of BPs studied in this paper, we are motivated to design an ISNN architecture that guarantees to output a supermodular approximation $\tilde{\phi}(\tilde{x})$ of the true value function $\phi(x)$, where we set $\tilde{x} := [x^\top, (\mathbf{1} - x)^\top]^\top$ in the case of ISNN. Long et al. (2023) and Chen et al. (2021) provide some sufficient conditions for parametric optimization $\phi(x)$ to be supermodular.

First, we recall the definition of supermodularity.

**Definition 2.** *Consider a subset $D$ of $\mathbb{R}^n$, a function $f : D \to \mathbb{R}$, and for $x, y \in D$ define $x \vee y = [\max\{x_1, y_1\}, \ldots, \max\{x_n, y_n\}]^\top$ and $x \wedge y = [\min\{x_1, y_1\}, \ldots, \min\{x_n, y_n\}]^\top$. Then, $f$ is called supermodular if $f(x) + f(y) \leq f(x \vee y) + f(x \wedge y)$ for all $x, y \in D$. In addition, a set $S$ is called a lattice if $x \vee y \in S$ and $x \wedge y \in S$ for all $x, y \in S$.*

We remark that this definition of supermodularity applies to a mixed-integer-valued argument and the domain $D$ of a supermodular function $f$ needs not to be $\{0, 1\}^n$.

Then, we are ready to present the main result of this section.

**Proposition 3.** *The function $\tilde{\phi}(\tilde{x})$ output from ISNN is supermodular in $\tilde{x}$ if $W_{1:(K+1)}$ and $D_{2:K}$ are non-negative, and $\sigma(\cdot)$ is convex and non-decreasing (e.g., ReLU).*

The proof is given in Appendix A.2. By Proposition 3, we simply need to add appropriate sign constraints to some weight parameters when training the ISNN.

Next, Proposition 4 shows the representability of ISNN.

**Proposition 4.** *Consider an arbitrary (not necessarily supermodular) $\phi : \{0, 1\}^n \to \mathbb{R}$. Then, the following statements hold.*
*i. (**Universal Approximation**) there exists an ISNN with the architecture in Figure 2 such that $\tilde{\phi}(x, \mathbf{1} - x) = \phi(x)$ for all $x \in \{0, 1\}^n$.*
*ii. (**Maximum Representability**) With a fixed number of neurons $N_{nr}$ in its hidden layers, the ISNN's architecture does not affect its representability (measured by the number of parameters the ISNN trains).*
*iii. (**Sufficient Fitting**) Given $N_s$ sample-label pairs, the ISNN is able to output an approximation $\tilde{\phi}$ that fits these pairs exactly if*

$$N_{nr} \geq \left\lceil \frac{N_s}{2n + 1} - 1 \right\rceil . \tag{8}$$

We give a proof of Proposition 4 in Appendix A.3. Since the ISNN's architecture does not affect its representability, to better compare with GNN, we incorporate two hidden layers in ISNN, each with $N_{nr}/2$ neurons, where $N_{nr}$ is the smallest integer that satisfies (8).

Thanks to the supermodularity of $\tilde{\phi}(\tilde{x})$ promised by Proposition 3, we can recast the approximate optimality condition $g(y, x) \geq \tilde{\phi}(\tilde{x})$ as linear inequalities, in lieu of linearizing $\tilde{\phi}(\tilde{x})$ using auxiliary binary variables and big-$M$ coefficients. We place the theoretical analysis for supermodular $\tilde{\phi}(\tilde{x})$ in Appendix B.

## 3.3 SOLVING

### 3.3.1 MIXED-INTEGER REPRESENTATION OF $\tilde{\phi}(\tilde{x})$

We represent the epigraph of $\tilde{\phi}(\tilde{x})$ as linear inequalities using auxiliary binary variables, to be embedded into the BP reformulation (2)–(3). To this end, we recall that the ReLU activation function takes the form

$$\sigma(x) = \max\{x, 0\} = \begin{cases} x, \ x \geq 0 \\ 0, \ x < 0. \end{cases} \tag{9}$$

Then, $\sigma(x)$ can be rewritten as

$$0 \leq \ \sigma(x) \ \leq M\delta \tag{10a}$$
$$x \leq \ \sigma(x) \ \leq x + M(1 - \delta) \tag{10b}$$

where $\delta$ is an auxiliary binary variable and $M$ is a sufficiently large positive number. We note that, in computation, the value of $M$ does not have to be arbitrarily big and it can be iteratively strengthened by solving relaxations of (2)–(3); see, e.g., Qiu et al. 2014. Likewise, the defition of $\tilde{\phi}(\tilde{x})$ in (6) can be represented layer by layer as follows:

$$\begin{cases} z'_1 = W_1\tilde{x} + b_1 \\ 0 \leq \ z_1 \ \leq M\delta_1 \\ z'_1 \leq \ z_1 \ \leq z'_1 + M(1 - \delta_1) \end{cases} \tag{11a}$$

$$\begin{cases} z'_k = W_kz_{k-1} + b_k + D_k\tilde{x} \\ 0 \leq \ z_k \ \leq M\delta_k \qquad \forall k = 2, ..., K \\ z'_k \leq \ z_k \ \leq z'_k + M(1 - \delta_k) \end{cases} \tag{11b}$$

$$\tilde{\phi} = W_{K+1}z_K + b_{K+1} + D_{K+1}\tilde{x}, \tag{11c}$$

where $z'_k$ and $\delta_k$ are auxiliary continuous and binary variables for all $1 \leq k \leq K$, respectively. Plugging this representation into (3) yields a mixed-integer programming approximation of the BP (1):

$$\min_{x,y} f(x,y) \tag{12a}$$

$$\text{s.t. } x \in X, \ y \in Y(x) \tag{12b}$$

$$g(y,x) \geq \tilde{\phi}(\tilde{x}), \ (11) \tag{12c}$$

$$\tilde{x} = \begin{cases} x & \text{if using GNN,} \\ [x^\top, (\mathbf{1} - x)^\top]^\top & \text{if using ISNN.} \end{cases} \tag{12d}$$

Here we note that existing algorithms and highly-efficient off-the-shelf solvers can be adopted to handle the single-level program (12), which is yet out of the scope of this paper.

### 3.3.2 Neural Bilevel Algorithm

Combining the above sampling, training, and solving processes, we conclude our neural bilevel algorithm in Algorithm 2. By iteratively conducting the enhanced sampling, we can find new and higher-quality samples and improve the accuracy of the found solution.

---

**Algorithm 2** Neural Bilevel Algorithm

---

1: **INPUT**: maximum number of iterations $N_{\text{iteration}}$, initial upper bound $f_{ub} := +\infty$.
2: Solve (2) and store an optimal solution $x^*$.
3: Store a $y^* \in \text{argmax}_{y \in Y(x^*)} \ g(y, x^*)$.
4: Update $f_{ub} \leftarrow \min\{f_{ub}, f(x^*, y^*)\}$.
5: **for** $i = 1, ..., N_{\text{iteration}}$ **do**
6:      Conduct Algorithm 1 to obtain the pairs $(\hat{x}, \phi(\hat{x}))$ and their corresponding $f(\hat{x}, \hat{y})$.
7:      Train a neural network $\tilde{\phi}(\tilde{x})$ (GNN or ISNN) using $(\hat{x}, \phi(\hat{x}))$.
8:      Solve (12) using the trained $\tilde{\phi}$ and store an optimal solution $x^*$.
9:      Store a $y^* \in \text{argmax}_{y \in Y(x^*)} \ g(y, x^*)$.
10:     Update $f_{ub} \leftarrow \min\{f_{ub}, f(x^*, y^*), f(\hat{x}, \hat{y})\}$.
11: **end for**
12: **OUTPUT**: Current (i.e., best) upper bound $f_{ub}$ and its corresponding solution $(x, y)$.

---

## 4 Experiments

### 4.1 Illustrative Examples

We first use an illustrative example to show the effectiveness of our proposed methods. We consider a BP with a nonconvex and nonlinear program in the lower level:

$$\min_{x} 2x_1 + x_2 - 3y$$
$$\text{s.t. } x_1, x_2 \in \{0, 1\} \tag{13a}$$
$$\text{where } y^* \in \arg\max \ -(y-2)^2$$
$$\text{s.t. } 0 \leq y \leq 1 + 2|x_1 - x_2| \tag{13b}$$

This BP admits the following optimality condition for the lower-level problem:

$$-(y-2)^2 \geq \phi(x_1, x_2). \tag{14}$$

where $\phi(x_1, x_2) := \max\{-(y-2)^2 : (13b)\}$. In this example, $\phi(x_1, x_2)$ admits a closed-form expression. Indeed, by the structure of the lower-level objective function, we notice that $y^* = \min\{1 + 2|x_1 - x_2|, 2\}$ and so $\phi(x_1, x_2) = -(\min\{1 + 2|x_1 - x_2|, 2\} - 2)^2$, which is depicted in Figure 3(a). Incorporation of this expression yields the optimal solution $(x_1^*, x_2^*) = (0, 1)$. In Figure 8(a) of Appendix D.1, we illustrate the feasible region of $(x_1, x_2, y)$ described by constraints (13a)–(13b) and why (14) ensures optimality.

On the other hand, following the steps and methods described in Section 3, we use neural networks to approximate the value function $\phi(x_1, x_2)$. Thanks to the simplicity of this example, we find closed-form expressions of these approximations: $\tilde{\phi}_G(x_1, x_2) = x_1 + x_2 - 1 - 2\sigma(x_1 + x_2 - 1)$ for GNN and $\tilde{\phi}_{IS}(x_1, x_2) = x_1 + (1 - x_2) - 2 + 2\sigma((1 - x_1) + x_2 - 1))$ for ISNN, which are

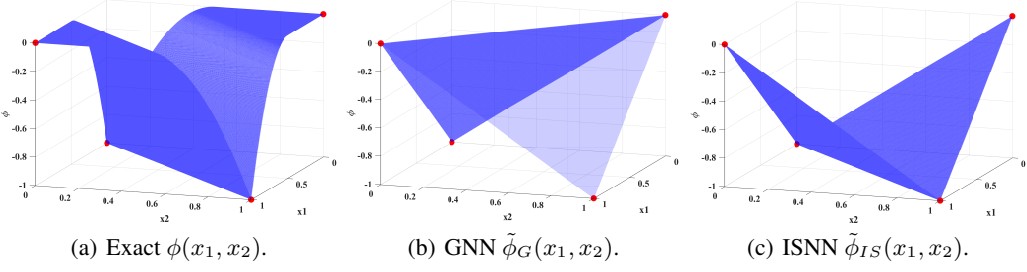

(a) Exact $\phi(x_1, x_2)$.     (b) GNN $\tilde{\phi}_G(x_1, x_2)$.     (c) ISNN $\tilde{\phi}_{IS}(x_1, x_2)$.

Figure 3: Value function.

depicted in Figures 3(b) and 3(c), respectively. Figures 8(b) and 8(c) in Appendix D.1 visualize their corresponding optimality cuts. Both approximations $\tilde{\phi}_G(x_1, x_2)$ and $\tilde{\phi}_{IS}(x_1, x_2)$ lead to the (true) optimal solution $x^* = (0, 1)^\top$. From these, we observe that incorporating the approximate $\tilde{\phi}(x_1, x_2)$ from either GNN or ISNN correctly produces the optimal solution.

## 4.2 RANDOMLY GENERATED INSTANCES

### 4.2.1 SETUP

We randomly generate 6 classes of instances for numerical experiments, with $n = 10, 20, 30, 40, 50, 60$. The details of instance generation is reported in Appendix C. We allow the lower-level problem to be a linear program (LP) or a mixed-integer linear program (MILP). By default, we generate 1,000 samples using Algorithm 1 and use them in training neural networks. We adopt ReLU as the activation function and design the architecture of neural networks by Proposition 1 and by Propositions 3–4 for GNN and ISNN, respectively. We use the Adam algorithm (Kingma & Ba, 2014) for training for 1000 epochs and set the learning rate as 0.001 with the decay 0.001.

### 4.2.2 COMPARISON WITH STATE-OF-THE-ART SOLVERS

We compare our method with the state-of-the-art solver for bilevel problems, MiBS (Tahernejad et al., 2020) and use its solution within a 1-hour time limit as the benchmark to calculate the objective differences (i.e., the gap between the objectives from our method and MiBS). The negative objective difference means that our method provides a better solution than the benchmark. For each instance, we replicate our method for 10 times, considering the randomness in sampling and training.

For the instances with a LP lower level, we report the objective difference of the best upper bound found by Algorithm 2 in Figure 4 and the average computational time in Figure 5. In all figures, the legend, for example, "GNN–2" means that we use GNN to fit samples and $N_{\text{iteration}} = 2$. Figure 4(a) and Figure 4(b) show the average and minimum objective difference in the 10 replications. The distribution of the objective difference of the 10 replications is provided in Figure 9 of Appendix D.2. The computational time reported in Figure 5 is the average time of the 10 replications. The itemized time in sampling, training, and solving is provided in Figure 10 of Appendix D.2.

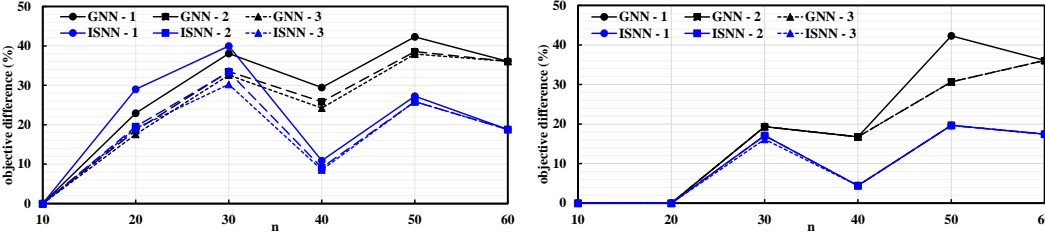

(a) Average objective difference of 10 replications.     (b) Minimum objective difference in 10 replications.

Figure 4: objective difference in different instances with linear lower level.

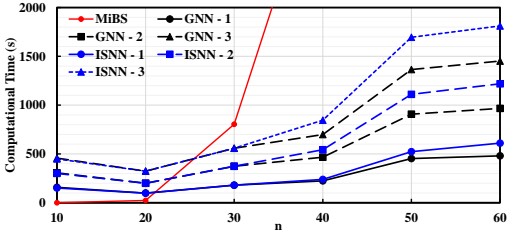

Figure 5: Computational time in different instances with a LP lower level.

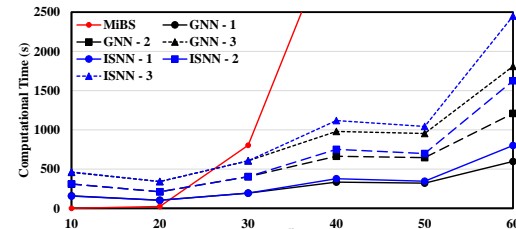

Figure 6: Computational time in different instances with a MILP lower level.

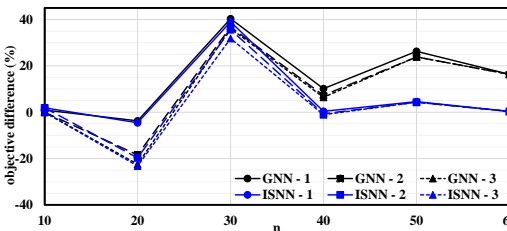

(a) Average objective difference of 10 replications.

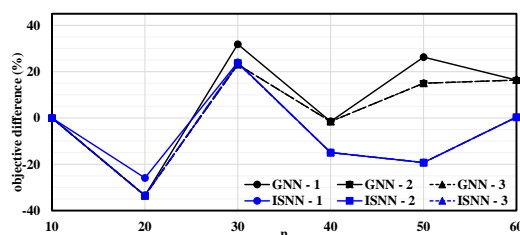

(b) Minimum objective difference of 10 replications.

Figure 7: objective difference in different instances with a MILP lower level.

From the results, we can see that when $n = 10$, both GNN and ISNN can produce a true optimal solution in all replications. This is because we can enumerate all feasible solutions $x$ in sampling for $n = 10$. Yet during the sampling process, there exists much repeated sampling, for which the computational time is slightly longer than that of $n = 20$. When $n \geq 20$, the computational time increases as $n$ gets larger, which mainly results from the longer sampling time, and is almost linear with $N_{\text{iteration}}$. However, the computational time is significantly shorter than that of MiBS when $n \geq 40$. In addition, the objective difference of instances with $n = 30, 50, 60$ is larger than 15%. The increase of $N_{\text{iteration}}$ results in the decrease of average objective difference for both GNN and ISNN. It means that the influence of sampling gets significant and validates that enhanced sampling helps reduce average objective difference. Yet as $n$ increases, the marginal improvement of enhanced sampling becomes smaller, which is caused by the increasing difficulty in finding high-quality samples. Comparing GNN and ISNN with the same $N_{\text{iteration}}$, we observe that ISNN always outperforms GNN in both average and minimum objective difference. Using ISNN can reduce objective difference by more than 10% when $n \geq 40$, which validates the effectiveness of ISNN in improving accuracy.

For the instances with a MILP lower level, we report the objective difference of the best upper bound found by Algorithm 2 in Figure 7 and the average computational time in Figure 6. More details are provided in Figures 11–12 of Appendix D.2. We observe that these results produce similar insights as those of the LP lower-level problems. Notably, our method has excellent accuracy in most instances (achieving objective difference less than 5%, only except when $n = 30$) and even outperforms MiBS when $n = 20, 40, 50$. This is because when the lower level is a MILP, it becomes significantly more challenging to find high-quality feasible solutions, and in these instances even MiBS reports incorrect optimal solutions. In contrast, the proposed method is able to produce better feasible solutions in (dramatically) shorter computational time, validating its effectiveness.

## 5 CONCLUSIONS

We considered machine learning methods for solving mixed-integer, nonconvex BPs with binary tender. We developed an enhanced sampling algorithm to find high-quality samples, designed GNN and ISNN to approximate the lower-level value function (in terms of the upper-level decisions), and incorporated the results as optimality cuts into a single-level reformulation of the BP. We validate the effectiveness of the proposed approaches using an illustrative example and larger-scale, randomly generated instances. Through these experiments, we demonstrated that the enhanced sampling helps reduce average objective difference and ISNN always outperforms GNN. The computational time of using either GNN or ISNN is significantly shorter than that of (a state-of-the-art solver) MiBS when the dimension of the biancy tender exceeds 30. Notably, in most instances with a MILP lower level, the proposed method can produce even better solutions than MiBS in shorter computational time.

ACKNOWLEDGMENTS

The authors would like to thank anonymous reviewers for their detailed and helpful comments. Ruiwei Jiang is supported in part by the U.S. Air Force Office of Scientific Research under the grant FA9550-23-1-0323.

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

# A    PROOFS

## A.1    PROOF OF PROPOSITION 1

Any arbitrary $\phi : \{0,1\}^n \to \mathbb{R}$ can be rewritten as

$$
\begin{aligned}
\phi(x) &= (1-x_1)(1-x_2)(1-x_3)\cdots(1-x_n)\phi(0,0,0,\ldots,0) \\
&\quad + x_1(1-x_2)(1-x_3)\cdots(1-x_n)\phi(1,0,0,\ldots,0) \\
&\quad + (1-x_1)x_2(1-x_3)\cdots(1-x_n)\phi(0,1,0,\ldots,0) \\
&\quad + x_1 x_2(1-x_3)\cdots(1-x_n)\phi(1,1,0,\ldots,0) \\
&\quad + (1-x_1)(1-x_2)x_3\cdots(1-x_n)\phi(0,0,1,\ldots,0) \\
&\quad + \cdots \\
&\quad + x_1 x_2 x_3 \cdots x_n \phi(1,1,1,\ldots,1) \\
&= \sum_{z \in \{0,1\}^n} \phi(z) \prod_{i=1}^{n} \Big[(1-z_i) + (2z_i - 1)x_i\Big].
\end{aligned}
$$

This shows that $\phi$ is a polynomial of (at most) degree $n$. Then, $\phi$ admits the following representation:

$$
\phi(x) \;=\; a\prod_{i} x_i + \sum_{\{j\}\subseteq[n]} b_j \prod_{i\neq j} x_i + \sum_{\{k,j\}\subseteq[n]} c_{k,j} \prod_{i\neq j,k} x_i + \cdots,
$$

where $a$, $b$, $c$, ... are coefficients of the polynomial. Since $x \in \{0,1\}^n$, we can rewrite

$$
\prod_{i\notin S} x_i \;=\; \max\Big\{\sum_{i\notin S} x_i - (n - |S| - 1), 0\Big\} \;=\; \sigma\Big(\sum_{i\notin S} x_i - (n - |S| - 1)\Big)
$$

for all subsets $S \subseteq [n]$. It follows that

$$
\phi(x) = a\sigma\left(\sum_{i} x_i - (n-1)\right) + \sum_{\{j\}\subseteq[n]} b_j \sigma\left(\sum_{i\neq j} x_i - (n-2)\right)
$$

$$
+ \sum_{\{k,j\}\subseteq[n]} c_{k,j} \sigma\left(\sum_{i\neq j,k} x_i - (n-3)\right) + \cdots.
$$

There are $2^n$ terms in this expression and the $n$ first-order terms and the zeroth-order term do not need activation. Therefore, $\phi(x)$ can be represented using a neural network with an architecture as in Figure 2 with $K = 1$ and $2^n - n - 1$ neurons (activations). This proves claim (i).

For a GNN as shown in Figure 2, $\tilde{\phi}(x)$ admits a closed-form expression as in (6). Let $z_k \in \mathbb{R}^{n_k}$ for all $k \in [K]$, then the number of neurons in the hidden layers is $N_{nr} = \sum_{k=1}^{K} n_k$ and the total number of parameters $N_{pm}$ GNN trains is

$$
\begin{aligned}
N_{pm} &= n_1(n+1) + \sum_{k=2}^{K} n_k(n_{k-1} + 1 + n) + (n_K + 1 + n) \\
&= \sum_{k=2}^{K} n_k n_{k-1} + n_K + (n+1)\Big(\sum_{k=1}^{K} n_k + 1\Big) \\
&= \sum_{k=2}^{K} n_k n_{k-1} + n_K + (n+1)(N_{nr} + 1).
\end{aligned}
$$

Then, with fixed $N_{nr}$, to maximize $N_{pm}$ by adjusting the GNN architecture boils down to optimizing $\sum_{k=2}^{K} n_k n_{k-1} + n_K$ over (integer) variables $K$ and $n_1, n_2, \ldots, n_K$. To this end, we state the following technical lemma, whose proof is relegated to Appendix A.1.1.

**Lemma 5.** *Consider the following (non-convex) quadratic program with continuous decision variables:*

$$\max_{n_1,\ldots,n_K} \quad \sum_{k=2}^{K} n_k n_{k-1} + n_K \tag{15a}$$

$$s.t. \quad \sum_{k=1}^{K} n_k = N_{nr}, \tag{15b}$$

$$n_k \geq 0 \quad \forall n \in [K]. \tag{15c}$$

*Then, an optimal solution to this program is $n_k = 0$ for all $k \in [K-2]$, $n_{K-1} = (N_{nr} - 1)/2$, and $n_K = (N_{nr} + 1)/2$. Additionally, the corresponding optimal value is $(N_{nr} + 1)^2/4$.*

By Lemma 5, the optimal solution to formulation (15) remains the same if $N_{nr}$ is odd and we restrict each $n_k$ to take integer values only, because $(N_{nr} - 1)/2, (N_{nr} + 1)/2 \in \mathbb{N}$.

In case that $N_{nr}$ is even, consider the (integer) solution $n_k = 0$ for all $k \in [K-2]$, $n_{K-1} = N_{nr}/2$, and $n_K = N_{nr}/2$. The objective value of this solution is $(N^2 + 2N)/4$, that is, $1/4$ less than the optimal value of (15). Since all coefficients of the objective function (15a) are integer-valued, the objective value of any (integer) solution must be integer-valued, too. It follows that there does not exist any other (integer) solution, whose objective value is strictly larger than $(N^2 + 2N)/4$. Therefore, the solution $n_k = 0$ for all $k \in [K-2]$, $n_{K-1} = N_{nr}/2$, and $N_K = N_{nr}/2$ is optimal when $N_{nr}$ is even. This proves claim (ii).

To fit the $N_s$ sample-label pairs exactly, we need the number of parameters $N_{pm}$ of the GNN to be at least as large as $N_s$, i.e., $N_{pm} \geq N_s$. But claim (ii) implies that, when choosing $n_{K-1} = n_K = N_{nr}/2$,

$$N_{pm} = \frac{1}{4}(N_{nr}^2 + 2N_{nr}) + (n+1)(N_{nr} + 1) \geq N_s.$$

It follows that

$$N_{nr} \geq \sqrt{(2n+2)^2 + 4N_s + 1} - (2n+3),$$

proving claim (iii).

### A.1.1  PROOF OF LEMMA 5

For each local optimal solution $n$ to formulation (15), the KKT condition states that there exist Lagrangian multipliers $\mu \in \mathbb{R}_+^K$ and $\lambda \in \mathbb{R}$, such that

$$\mu_k n_k = 0 \quad \forall k \in [K], \tag{16a}$$

$$\mu_k - \lambda + n_{k-1} + n_{k+1} = 0 \quad \forall k \in [K], \tag{16b}$$

where $n_0 := 0$ and $n_{K+1} := 1$. For example, for the solution $n^*$ with $n_k^* = 0$ for all $k \in [K-2]$, $n_{K-1}^* = (N_{nr} - 1)/2$, and $n_K^* = (N_{nr} + 1)/2$, we pair it with the Lagrangian multipliers $\lambda^* = (N_{nr} + 1)/2$ and

$$\mu_k^* = \begin{cases} \frac{1}{2}(N_{nr} + 1) & \text{if } k \in [K-3] \\ 1 & \text{if } k = K-2 \\ 0 & \text{if } k = K-1, K \end{cases} \quad \forall k \in [K].$$

Since $(n^*, \mu^*, \lambda^*)$ satisfies (16a)–(16b) as well as (15b)–(15c), $n^*$ is a local optimal solution to (15) with objective value $(N_{nr} + 1)^2/4$. In what follows, we shall show that any other local optimal solution to (15) does not achieve a strictly larger objective value, establishing the global optimality of $n^*$.

To this end, we multiply both sides of (16b) by $n_k$ and then sum them up for all $k \in [K]$ to yield

$$\sum_{k=1}^{K} \mu_k n_k - \lambda \sum_{k=1}^{K} n_k + 2 \sum_{k=2}^{K} n_k n_{k-1} + n_K = 0$$

$$\implies \sum_{k=2}^{K} n_k n_{k-1} + n_K = \frac{1}{2}(N_{nr}\lambda + n_K),$$

which follows from (16a) and (15b). This rewrites the quadratic program (15) as

$$\max_{n\geq 0,\lambda\geq 0,\mu\in\mathbb{R}}\ \frac{1}{2}(N_{nr}\lambda + n_K)$$

$$\text{s.t. } (15b)–(15c), (16a)–(16b).$$

For any local optimal solution $n$, together with its corresponding Lagrangian multipliers $(\mu, \lambda)$ satisfying (15b)–(15c) and (16a)–(16b), we discuss the following three cases.

1. If $\mu_K = 0$, then (16b) with $k = K$ implies that

$$0 = n_{K-1} + 1 + \mu_K - \lambda = n_{K-1} + 1 - \lambda \quad \Longrightarrow \quad \lambda = n_{K-1} + 1.$$

We denote $m := \sum_{k=1}^{K-2} n_k$ with $0 \leq m \leq N_{nr}$. The following two sub-cases show that $n$ cannot be better than $n^*$.

(a) If $n_{K-1} = 0$, then $\lambda = 1$ and $n_K = N_{nr} - \sum_{k=1}^{K-1} n_k = N_{nr} - m$. It follows that the obejctive value of $n$ is

$$\frac{1}{2}(N_{nr}\lambda + n_K) = N_{nr} - \frac{1}{2}m \leq N_{nr} \leq \frac{(N_{nr} + 1)^2}{4},$$

that is, smaller than that of $n^*$. This finishes the proof.

(b) If $n_{K-1} > 0$, then $\mu_{K-1} = 0$ by (16a). It follows from (16b) with $k = K - 1$ that

$$0 = n_{K-2} + n_K - \mu_{K-1} - \lambda = n_{K-2} + n_K - n_{K-1} - 1 \quad \Longrightarrow \quad n_{K-1} - n_K = n_{K-2} - 1.$$

But $n_{K-1} + n_K = N - m$ by definition of $m$. This implies that

$$n_{K-1} = \frac{1}{2}(N_{nr} - m + n_{K-2} - 1) \quad \text{and} \quad n_K = \frac{1}{2}(N_{nr} - m - n_{K-2} + 1).$$

Then, the objective value of $n$ is

$$\begin{aligned}
\frac{1}{2}(N_{nr}\lambda + n_K) &= \frac{1}{2}N_{nr}(n_{K-1} + 1) + \frac{1}{2}n_K \\
&= \frac{1}{4}\left[N_{nr}^2 + (n_{K-2} + 2 - m)N_{nr} + (1 - n_{K-2} - m)\right] \\
&\leq \frac{1}{4}\left[N_{nr}^2 + 2N_{nr} + (1 - 2m)\right] \\
&\leq \frac{1}{4}(N_{nr} + 1)^2,
\end{aligned}$$

where the first inequality is because, with fixed $m$, setting $n_{K-2} = m$ maximizes the objective value. Hence, the objective value of $n$ is smaller than that of $n^*$, finishing the proof.

2. If $\mu_K > 0$ and $\mu_{K-1} = 0$, then $n_K = 0$ by (16a) and so (16b) implies that, with $k = K-1$,

$$0 = \mu_{K-1} - \lambda + n_{K-2} + n_K = -\lambda + n_{K-2},$$

that is, $\lambda = n_{K-2}$. The following two sub-cases show that $n$ cannot be better than $n^*$.

(a) If $\mu_{K-2} > 0$, then $\lambda = n_{K-2} = 0$ by (16a). Hence, the objective value of $n$ is $(N_{nr}\lambda + n_K)/2 = 0$, smaller than that of $n^*$. This finishes the proof.

(b) If $\mu_{K-2} = 0$, then (16b) with $k = K - 2$ implies that

$$0 = \mu_{K-2} + n_{K-3} + n_{K-1} + \lambda = n_{K-3} + n_{K-1} + \lambda,$$

that is, $\lambda = n_{K-3} + n_{K-1}$. But (15b) implies that

$$N_{nr} \geq n_{K-3} + n_{K-2} + n_{K-1} = 2\lambda \implies \lambda \leq \frac{N_{nr}}{2}.$$

It follows that the objective value of $n$ is

$$\frac{1}{2}(N_{nr}\lambda + n_K) \leq \frac{N_{nr}^2}{4} \leq \frac{(N_{nr} + 1)^2}{4},$$

that is, smaller than that of $n^*$. This finishes the proof.

3. If $\mu_K > 0$ and $\mu_{K-1} > 0$, then $n_K = n_{K-1} = 0$ by (16a). Let $j \in [K-2]$ denote the largest index such that $n_j > 0$. Then, $\mu_j = 0$ by (16a). We discuss the following three sub-cases about $j$ to show that $n$ cannot be better than $n^*$.

   (a) If $j = 1$, then $\mu_1 = 0$ and $n_2 = 0$ by definition of $j$. It follows from (16b) with $k = 1$ that

   $$0 = n_2 + \mu_1 - \lambda = -\lambda \implies \lambda = 0.$$

   Hence, the objective value of $n$ is $(N_{nr}\lambda + n_K)/2 = 0$, smaller than that of $n^*$. This finishes the proof.

   (b) If $j \geq 2$ and $n_{j-1} > 0$, then $\mu_{j-1} = \mu_j = 0$ by definition of $j$. It follows from (16b) with $k = j - 1$ and $k = j$ that

   $$\mu_{j-1} - \lambda + n_{j-2} + n_j = 0 \implies \lambda = n_{j-2} + n_j,$$
   $$\mu_j - \lambda + n_{j-1} + n_{j+1} = 0 \implies \lambda = n_{j-1} + n_{j+1}.$$

   But (15b) implies that

   $$N_{nr} \geq n_{j-2} + n_{j-1} + n_j + n_{j+1} = 2\lambda \implies \lambda \leq \frac{N_{nr}}{2}.$$

   It follows that the objective value of $n$ is

   $$\frac{1}{2}(N_{nr}\lambda + n_K) \leq \frac{N_{nr}^2}{4} \leq \frac{(N_{nr}+1)^2}{4},$$

   that is, smaller than that of $n^*$. This finishes the proof.

   (c) If $j \geq 2$ and $n_{j-1} = 0$, then (16b) with $k = j$ implies that

   $$0 = \mu_j - \lambda + n_{j-1} + n_{j+1} = -\lambda \implies \lambda = 0,$$

   where the second equality follows from the definition of $j$. Hence, the objective value of $n$ is $(N_{nr}\lambda + n_K)/2 = 0$, smaller than that of $n^*$. This finishes the proof.

## A.2 PROOF OF PROPOSITION 3.

Consider functions $f : \mathbb{R} \to \mathbb{R}$, $g : \mathbb{R}^n \to \mathbb{R}$, and their composite $f(g(x))$. It can be shown that $f(g(x))$ is supermodular in $x$ if one of the following is satisfied (see, e.g., Proposition 2.2.5(c) in Simchi-Levi et al. (2005)):

1. $f$ is increasing and convex, and $g$ is increasing and supermodular;

2. $f$ is decreasing and convex, and $g$ is increasing and submodular.

Here, a function $g$ is called increasing if $g(x) \leq g(x')$ whenever $x \leq x'$, decreasing if $-g$ is increasing, and submodular if $-g$ is supermodular.

We proceed to prove by induction. For each $k \leq K$, suppose that $z_{k-1}$ is increasing and supermodular in $x$. Then, $W_k z_{k-1} + b_k + D_k x$ is increasing and supermodular in $x$ because $W_k \geq 0$ and $D_k \geq 0$. It follows that $z_k$ is increasing and supermodular in $x$ because $\sigma$ is increasing and convex. In particular, $z_K$ is increasing and supermodular in $x$.

Finally, $\tilde{\phi}$ is supermodular in $x$ by (6c) because $W_{K+1} \geq 0$ and $D_{K+1}x$ is supermodular (actually linear) in $x$. This finishes the proof.

### A.3 PROOF OF PROPOSITION 4.

Define $x_i' := 1 - x_i$ for all $i \in [n]$ and $m_\phi := \min_{z \in \{0,1\}^n} \phi(z)$. Then, any arbitrary $\phi : \{0,1\}^n \to \mathbb{R}$ can be rewritten as

$$
\begin{aligned}
\phi(x) &= \sum_{z \in \{0,1\}^n} \phi(z) \prod_{i=1}^{n} \left[ z_i x_i + (1 - z_i) x_i' \right] \\
&= \sum_{z \in \{0,1\}^n} \phi(z) \max \left\{ \sum_{i=1}^{n} \left[ z_i x_i + (1 - z_i) x_i' \right] - (n-1), \, 0 \right\} \\
&= \sum_{z \in \{0,1\}^n} \phi(z) \, \sigma \left( \sum_{i=1}^{n} \left[ z_i x_i + (1 - z_i) x_i' \right] - (n-1) \right) \\
&= m_\phi + \sum_{z \in \{0,1\}^n} \left( \phi(z) - m_\phi \right) \sigma \left( \sum_{i=1}^{n} \left[ z_i x_i + (1 - z_i) x_i' \right] - (n-1) \right),
\end{aligned}
$$

where the second equality is because all $x_i$ and $x_i'$ in the product are binary, the third equality is by the definition of the activation function, and the last equality is because $\sigma \left( \sum_{i=1}^{n} \left[ z_i x_i + (1 - z_i) x_i' \right] - (n-1) \right) = 1$ if and only if $z = x$. Then, the coefficients of all $x_i$ and $x_i'$ are nonnegative (because $z_i \geq 0$ and $1 - z_i \geq 0$), and the coefficients of all activation functions are $\phi(z) - m_\phi \geq 0$. It follows that $\phi(x)$ can be represented using an ISNN with an architecture as in Figure 2 with $K = 1$ hidden layer, which consists of $2^n$ neurons and nonnegative coefficients only. This proves claim (i).

For an ISNN as shown in Figure 2, $\tilde{\phi}(\tilde{x})$ admits a closed form expression as in (6). Since $W_{1:K+1}$ and $D_{2:K}$ are nonnegative, the expression of $z_{k,i}$, the $i$th entry of $z_k$, in ISNN can be rewritten as

$$
\begin{aligned}
z_{k,i} &= \sigma \left( \sum_{j} W_{k,ij} z_{k-1,j} + b_{k,i} + D_{k,i} \tilde{x} \right) \\
&= \sigma \left( \sum_{j} W_{k,ij} \sigma(W_{k-1,j} z_{k-2} + b_{k-1,j} + D_{k-1,j} \tilde{x}) + b_{k,i} + D_{k,i} \tilde{x} \right) \\
&= \sigma \left( \sum_{j} \sigma(W_{k,ij} W_{k-1,j} z_{k-2} + W_{k,ij} b_{k-1,j} + W_{k,ij} D_{k-1,j} \tilde{x}) + b_{k,i} + D_{k,i} \tilde{x} \right) \\
&= \sigma \left( \sum_{j} \sigma(W_{k-1,j}' z_{k-2} + b_{k-1,j}' + D_{k-1,j}' \tilde{x}) + b_{k,i} + D_{k,i} \tilde{x} \right),
\end{aligned}
$$

where $W_{k,ij}$ denotes the entry of $W_k$ in row $i$ and column $j$, $W_{k-1,j}$ denotes the $j$th row of $W_{k-1}$, and

$$
W_{k-1,j}' := W_{k,ij} W_{k-1,j} \geq 0, \quad b_{k-1,j}' := W_{k,ij} b_{k-1,j}, \quad D_{k-1,j}' := W_{k,ij} D_{k-1,j} \geq 0.
$$

That is, the same $z_{k,i}$ can be represented by an ISNN with $W_{k,ij} = 1$ for all $i, j$, i.e., $W_k$ is an all-one matrix. In light of this, the ISNN admits the following equivalent expression:

$$
\begin{aligned}
z_1 &= \sigma(W_1 \tilde{x} + b_1) \\
z_k &= \sigma(\mathbf{1} z_{k-1} + b_k + D_k \tilde{x}), \ k = 2, ..., K \\
\tilde{\phi} &= \mathbf{1}^\top z_K + b_{K+1} + D_{K+1} \tilde{x},
\end{aligned}
$$

where $\mathbf{1}$ denotes an all-one vector or all-one matrix. Hence, the total number of parameters $N_{pm}$ to be trained in the ISNN is

$$N_{pm} = n_1(2n+1) + \sum_{k=2}^{K} n_k(2n+1) + (2n+1)$$
$$= \left( \sum_{k=1}^{K} n_k + 1 \right)(2n+1)$$
$$= (N_{nr}+1)(2n+1),$$

where $z_k \in \mathbb{R}^{n_k}$ for all $k \in [K]$ and so the number of neurons in the ISNN is $N_{nr} = \sum_{k=1}^{K} n_k$. Therefore, for fixed $N_{nr}$, $N_{pm}$ is independent of $W$ and $K$. This proves claim (ii).

To fit the $N_s$ sample-label pairs exactly, we need the number of parameters $N_{pm}$ of the ISNN to be at least as large as $N_s$, i.e.,

$$N_{pm} = (N_{nr}+1)(2n+1) \geq N_s.$$

It follows that

$$N_{nr} \geq \frac{N_s}{2n+1} - 1,$$

proving claim (iii).

### A.4 Approximation Error of $\tilde{\phi}$

We evaluate the error of using the $\tilde{\phi}$ obtained from neural networks (GNN or ISNN) to approximate the true (but unknown) value function $\phi$, defined as $\|\phi - \tilde{\phi}\|_\infty := \max_{x \in X} |\phi(x) - \tilde{\phi}(x)|$. The following proposition provides an upper bound of the approximation error, as a function of the lower-level problem and the neural network.

**Proposition 6.** *Consider a mixed-integer and linear lower-level problem, i.e., $g(y,x) := g^\top y$ and $Y(x) := \{y \in \mathbb{R}_+^{n-p} \times \mathbb{Z}_+^p : Gy = Tx + h\}$, where $g \in \mathbb{R}^m$, $G \in \mathbb{R}^{q \times m}$, $T \in \mathbb{R}^{q \times n}$, and $h \in \mathbb{R}^q$. Then, it holds that*

$$\|\phi - \tilde{\phi}\|_\infty \leq (C\|g\|_2 + L)d,$$

*where $C$ is a constant depending only on $n$ and $G$, $L$ is a constant depending only on the neural network coefficients $\{W_k\}_{k=1}^{K+1}$ and $\{D_k\}_{k=2}^{K+1}$, and $d$ is the Hausdorff distance between the sets of sampled and unsampled points in $X$, i.e.,*

$$d := \max_{x \in X \setminus \Omega} \min_{z \in \Omega} \|x - z\|_2.$$

*Proof:* Pick any $x, x' \in X$. First, using the representation (6) of the neural network, we have $\|z_1 - z_1'\|_2 \leq \|W_1\|_2(x - x')$, and

$$\|z_k - z_k'\|_2 \leq \|W_k\|_2 \|z_{k-1} - z_{k-1}'\|_2 + \|D_k\|_2 \|x - x'\|_2$$

for all $k = 2, \ldots, K$. A mathematical induction on $k$ yields that

$$|\tilde{\phi}(x) - \tilde{\phi}(x')| \leq L\|x - x'\|_2,$$

where $L = \prod_{k=1}^{K+1} \|W_k\|_2 + \sum_{k=2}^{K+1} \|D_k\|_2 (\prod_{\ell=k+1}^{K+1} \|W_\ell\|_2)$.

Second, since $\phi(x)$ represents the optimal value of a mixed-integer linear program parameterized by $x$, by Cook et al. (1986) we have

$$|\phi(x) - \phi(x')| \leq C_1 \|g\|_2 \|x - x'\|_2 + C_2 \|g\|_2,$$

where $C_1, C_2$ are constants that depend only on $n$ and $G$. But since $x, x' \in \{0,1\}^n$, $\phi(x) - \phi(x') = 0$ when $x = x'$ and $\|x - x'\|_2 \geq 1$ when $x \neq x'$. It follows that

$$|\phi(x) - \phi(x')| \leq C_1 \|g\|_2 \|x - x'\|_2 + C_2 \|g\|_2 \|x - x'\|_2$$
$$= C \|g\|_2 \|x - x'\|_2,$$

where $C = C_1 + C_2$.

Third, pick a $z \in \operatorname{argmin}_{z \in \Omega} \|x - z\|_2$. Then,

$$
\begin{aligned}
|\phi(x) - \tilde{\phi}(x)| &= |\phi(x) - \phi(z) + \tilde{\phi}(z) - \tilde{\phi}(x)| \\
&\leq |\phi(x) - \phi(z)| + |\tilde{\phi}(z) - \tilde{\phi}(x)| \\
&\leq C \|g\|_2 \|x - z\|_2 + L \|x - z\|_2 \\
&= (C\|g\|_2 + L)\|x - z\|_2,
\end{aligned}
$$

where the first equality is because $z \in \Omega$. Since $x$ is arbitrary, taking the maximum over all $x \in X$ of both sides of the inequality finishes the proof.

## B  EFFICIENT CALCULATION FOR ISNN

Although Section 3 provides a framework to solve general BPs through value function approximation, if there are total $N_{nr}$ neurons in the neural network, we need to introduce $N_{nr}$ auxiliary binary variables $\delta$, $N_{nr}$ auxiliary continuous variables $z'$, and $4N_{nr}$ big-$M$ constraints (see (11)) to represent $\tilde{\phi}(\tilde{x})$, which weakens its scalability.

Thanks to the supermodularity of $\tilde{\phi}(\tilde{x})$ promised by Proposition 3, we can replace (12c) with linear inequalities without introducing auxiliary binary variables or big-$M$ constraints as in (11). To this end, we define set $[2n] := \{1, 2, \ldots, 2n\}$ and the indicator set of a binary vector $\tilde{x}$ as $S(\tilde{x}) := \{1 \leq k \leq 2n : \tilde{x}_k = 1\}$. In addition, we define a set function $\varphi : 2^{[2n]} \to \mathbb{R}$ such that $\varphi(S(\tilde{x})) := \tilde{\phi}(\tilde{x})$ for all $\tilde{x} \in \{0, 1\}^{2n}$. Then, the following proposition recasts the (nonlinear) constraint $g(y, x) \geq \tilde{\phi}(\tilde{x})$ as linear inequalities.

**Proposition 7.** *(Adapted from Theorem 6 of Nemhauser & Wolsey (1981)). If $\tilde{\phi}(\tilde{x})$ is supermodular in $\tilde{x}$, then for all $\tilde{x} \in \{0, 1\}^{2n}$, $g(\cdot) \geq \tilde{\phi}(\tilde{x})$ if and only if*

$$
\begin{aligned}
g(\cdot) \geq{} & \varphi(S) - \sum_{k \in S} \rho([2n]\backslash\{k\}, k)(1 - \tilde{x}_k) \\
& + \sum_{k \in [2n]\backslash S} \rho(S, k)\tilde{x}_k, \qquad \forall S \subseteq [2n]
\end{aligned}
\tag{17}
$$

*where $\rho(S, k) := \varphi(S \cup \{k\}) - \varphi(S)$ for all $S \subseteq [2n]$ and $k \in [2n]\backslash S$.*

As compared to the reformulation (11), (17) does not introduce any new auxiliary variables. Yet, it involves an exponential number of linear inequalities which can nevertheless increase the formulation size. Fortunately, the inequalities (17) can be easily separated, that is, given fixed $\hat{x}$ and other variables, in polynomial time one can certify that $g(\cdot) \geq \tilde{\phi}(\hat{x})$ or find an violated inequality (17) with respect to some $S$. This is because a worst-case $S^*$ on the right-hand side of (17), i.e.,

$$
\begin{aligned}
S^* \in \operatorname*{argmax}_{S \subseteq [2n]} \Big\{ & \varphi(S) - \sum_{k \in S} \rho([2n]\backslash\{k\}, k)(1 - \hat{x}_k) \\
& + \sum_{k \in [2n]\backslash S} \rho(S, k)\hat{x}_k \Big\}
\end{aligned}
$$

admits a closed-form solution $S^* = S(\hat{x})$ (see Ljubić and Moreno (2018) and Qi et al. (2022)). Consequently, there is no need to incorporate the exponentially many inequalities (17) up front, and we only need to incorporate the violated ones on-the-fly, e.g., in a branch-and-bound algorithm for solving the BP.

## C  INSTANCE GENERATION

Following the instance generation rules of (Tahernejad et al., 2020), all instances are generated in the following forms:

$$\min_x c^{\mathrm{T}}x + d_1^{\mathrm{T}}y$$
$$\text{s.t. } A_1 x \le b_1$$
$$x \in \{0,1\}^n$$
$$y \in \arg\max_y d_2^{\mathrm{T}}y$$
$$\text{s.t. } A_2 x + B_2 y \le b_2$$
$$0 \le y \le \overline{y}$$
$$y \in R^m \text{ (or } y \in Z^m),$$

where $c$ is a $n \times 1$ vector, $d_1$ and $d_2$ are $m \times 1$ vectors. The constraint number is set to be the same as the variable number of each level, and hence, $A_1$ is a $n \times n$ matrix, $b_1$ is a $n \times 1$ vector, $A_2$ is a $m \times n$ matrix, $B_2$ is a $m \times m$ matrix, and $b_2$ is a $m \times 1$ vector. For instances with a linear lower-level problem, $y$ is continuous variables; for instances with a mixed-integer linear lower-level problem, $y$ is integer variables. In our numerical experiments, we set $m = 20, \overline{y} = 1$, and generate 6 instances with $n = 10, 20, 30, 40, 50, 60$. The coefficients are randomly generated in a range given in the following table, where $\delta = 200/(m + n)$ and $d_1 = d_2$.

Table 1: Range of randomly generated coefficients.

| COEFFICIENT | RANGE | COEFFICIENT | RANGE |
|---|---|---|---|
| $c, d_1, d_2$ | $[-50,\ 50]$ | $A_2$ | $[-10\delta,\ 10\delta]$ |
| $A_1$ | $[-2\delta,\ 2\delta]$ | $B_2$ | $[-\delta,\ \delta]$ |
| $b_1$ | $[30,\ 130]$ | $b_2$ | $[10,\ 110]$ |

# D  SUPPLEMENTARY RESULTS

## D.1  OPTIMALITY CUT IN ILLUSTRATIVE EXAMPLE

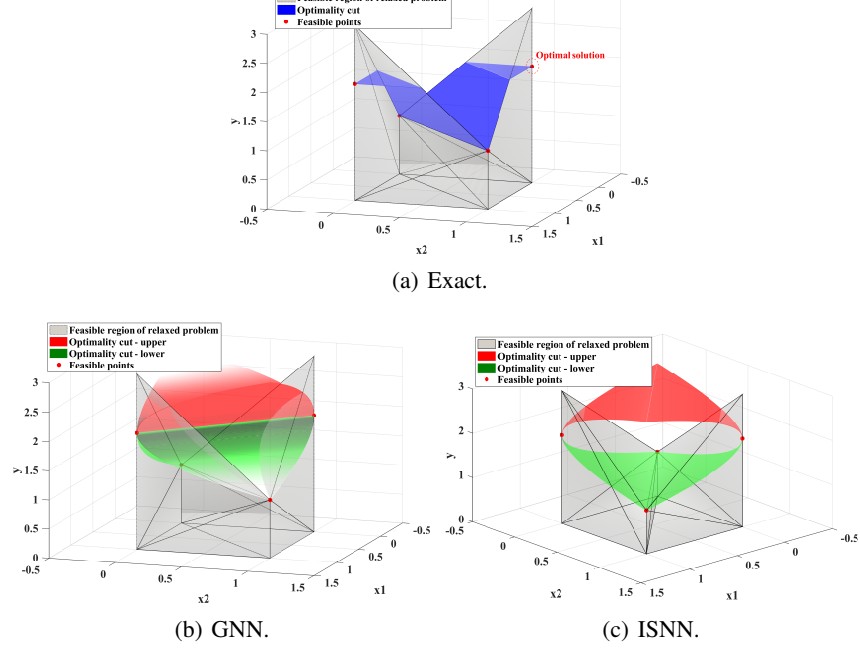

(a) Exact.

(b) GNN.

(c) ISNN.

Figure 8: Optimality cut.

## D.2 DETAILS IN GENERATED INSTANCES

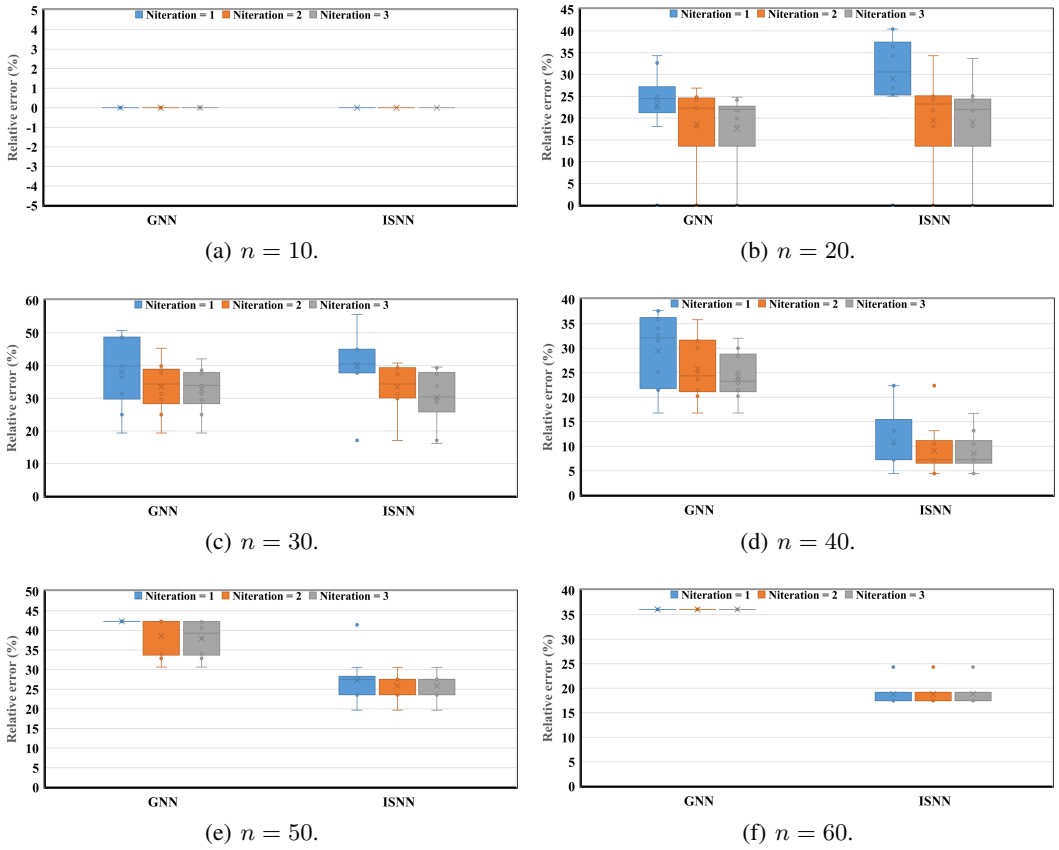

Figure 9: Distribution of objective difference in different instances with linear lower level.

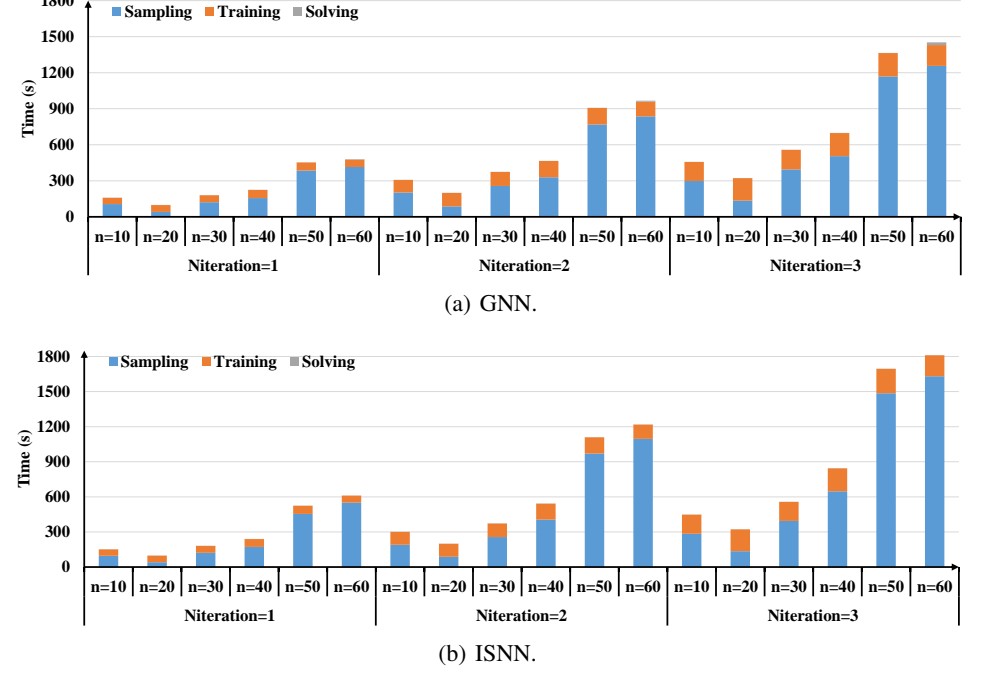

Figure 10: Itemized time in different instances with linear lower level.

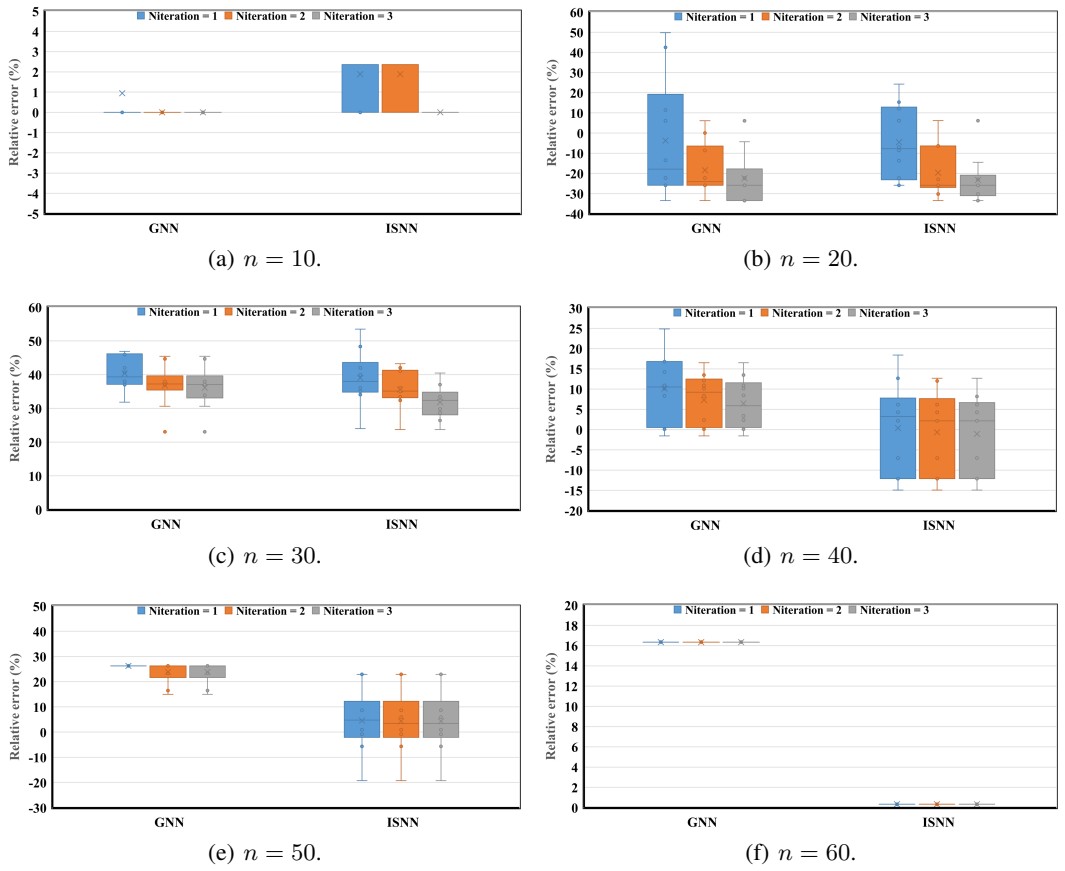

Figure 11: Distribution of objective difference in different instances with mixed-integer linear lower level.

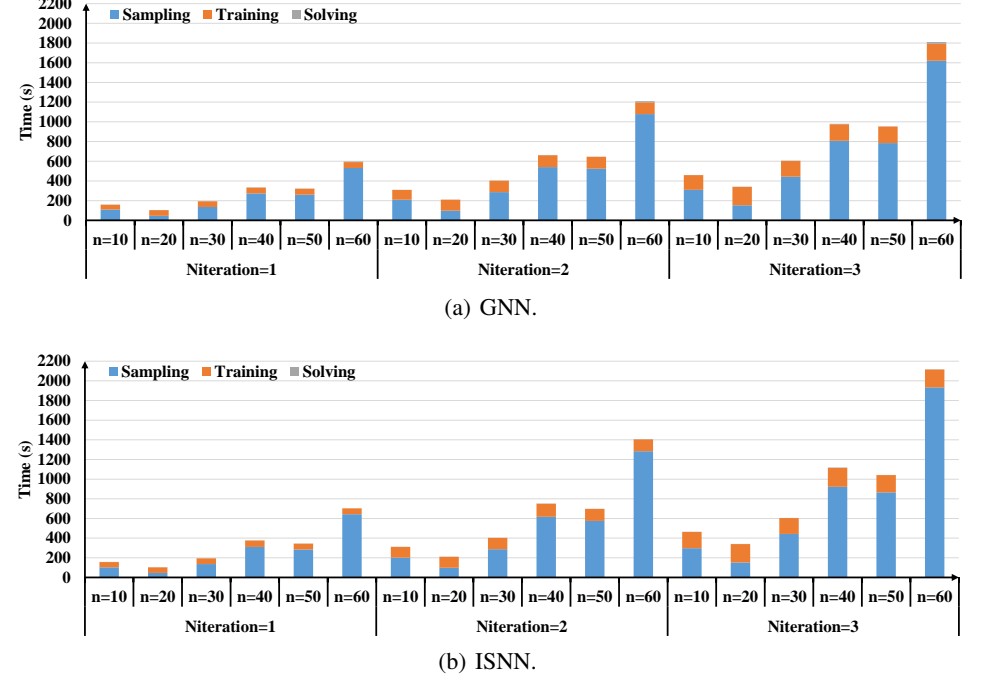

Figure 12: Itemized time in different instances with mixed-integer linear lower level.

### D.3 COMPARISON OF SAMPLING METHODS

We compare the efficiency of three sampling methods, including random sampling, Latin hypercube sampling (LHS), and the proposed enhanced sampling. We set a fixed sampling time of 30 minutes and adopt the generated MILP instances for numerical experiments. We report the number of obtained samples in Figure 13. We note that when $n = 10$, the maximum possible number of samples is 1024, for which enumeration can be adopted to find all feasible $x$. Yet when $n \geq 20$, enumeration becomes impractical and sampling is required.

From Figure 13, we see that the performance of random sampling is unstable and strongly depends on the lower-level problem. When the lower-level problem has a relatively large feasible region, such as the instance $n = 20$ or $n = 60$, it is easy for random sampling to find a feasible $x$, for which sampling efficiency is high. Yet if the lower-level problem has a relatively small feasible region, such as the instance $n = 30$ or $n = 40$, the sampling efficiency of random sampling gets quite low and can not find enough samples for training. In addition, because we do not have probability information, LHS has a similar performance to random sampling in most instances. However, through the proposed enhanced sampling, we guarantee to find a feasible $x$ in each sampling, and from Figure 13, the number of obtained samples is almost linearly decreasing with $n$. It is reasonable because a larger $n$ causes a longer time consumption for each sampling and thus fewer samples under a fixed sampling time. Therefore, to avoid the unstable performance of naive random sampling or other non-trivial sampling, we adopt the proposed enhanced sampling for all instances.

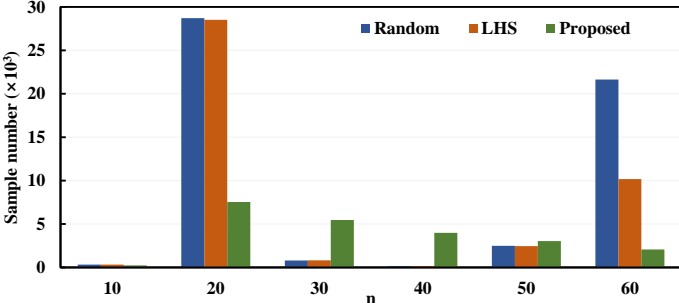

Figure 13: Neural network architecture.

### D.4 HIGHER DIMENSIONAL INSTANCES

We generate some larger instances with $n$ up to 120. We consider a MILP lower level and conduct experiments with a similar setup to Section 4.2.1. Figure 14 shows the numerical results.

From Figure 14(a), the average objective difference tends to increase as $n$ gets larger. When $n \geq 100$, the average objective difference gets higher than 5%. The results are reasonable because more samples are required to approximate $\phi(x)$ when $n$ gets larger yet our sample number is fixed. From Figure 14(b), the computational time increases as $n$ gets larger, and when $n \geq 100$, the computational time is longer than the set time limit of MiBS. Similar to Figure 12, sampling consumes the most computational time, for which more advanced sampling is required in future works.

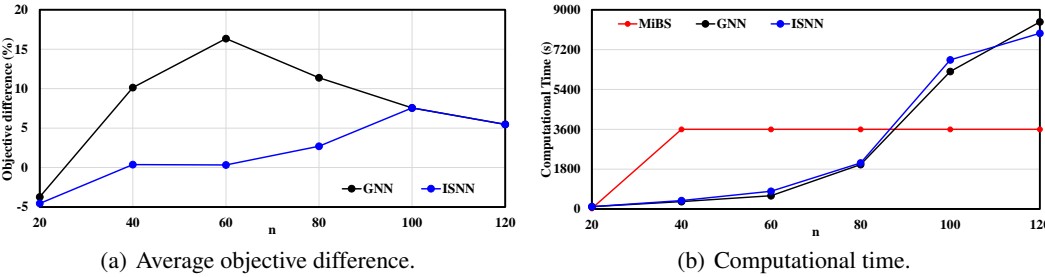

(a) Average objective difference.      (b) Computational time.

Figure 14: Instances with a MILP lower level.

