# OpenReview forum: "Learning to Solve Bilevel Programs with Binary Tender"
_ICLR.cc/2024/Conference — ICLR 2024 poster_

### Official Review · Reviewer_YfM6 · 2023-10-30

**Soundness:** 3 good
**Presentation:** 3 good
**Contribution:** 2 fair
**Rating:** 3
**Confidence:** 4

**Summary:**

The authors explore the use of machine learning techniques to solve complex bilevel programs with binary variables. They introduce a sampling algorithm for obtaining high-quality sample data for train neural networks. Two neural networks ( general neural networks (GNN) and input supermodular neural network  (ISNN) are proposed to estimate the lower-level value function based on upper-level decisions. These estimates are used to enhance the optimization process by adding optimality cuts to a single-level problem.

**Strengths:**

- The authors propose a machine learning-based algorithm for solving bilevel programs with binary variables in the outer level problem.

- The authors are able to provide some theoretical analysis for proposed neural networks with binary inputs.

- The paper is easy to follow.

**Weaknesses:**

1) The numerical experiments lack sufficient evidence to demonstrate the advantages of the proposed algorithm. It appears that the algorithm can only achieve optimality for test instances with n = 10. The authors solely conduct comparisons with the baseline method MiBS based on computational times, without assessing solution quality, such as the optimality gap.

2) The test instances are only from the authors’ synthetic generated ones. Some public datasets like MIBLP-XU and IBLP-FIS in Tahernejad et al., 2020 should be used.

3) This method should only be suitable when functions $f$ and $g$ are linear or MILP-representable because one needs to solve Eq (5) and (12) to optimality.

4) The cost to solve a single instance of Problem (5) is very expensive because it is at least as hard as a mixed-integer quadratic program (MIQP).

**Questions:**

1) What is the stopping criterion for Algorithm 2 so that the generated point $(x,y)$ is guaranteed to be feasible to Problem 1? That is, $x,y$ is satisfied $x \in X(y), y \in Y(x), g(y,x) \ge \max_{z \in Y(x)}  g(z,x)$

2) What is the definition of ``relative error” in Figures 4 and 7? How do you define an error when an optimal solution cannot be found, for example, for large-scale problems with n = 60?

3) Why relative errors can be negative (as in Fig. 7)?

4) Why we need to care of supermodularity of an ISNN? I notice that ISNN requires more neurons compared to GNN (i.e., the lower bound value for $N_{nr}$ in Eq. (8) vs Eq. (7))

5) Do we have to solve $\max_{y \in Y(\hat{x})} g(y, \hat{x})$ to optimality in Algorithm 1?

6) For feasibility purpose, why do we need the second term $h^T x$ in Eq. (5)?

7) I don’t fully understand the notation ``GNN representability” in Prop 1 (ii). This statement “we measure the GNN representability by the number of parameters it trains” seems to be not mathematically sound.

8) The architecture details of GNN of ISNN should be reported.

9) The name of section 3.3 seems to be incomplete.

10) I don’t see any part of Figures to support the claims “achieving relative error less than 5%,”, “the relative error of instances with n = 30, 50, 60 is larger than 15%.”

---

> ### Author Response · Authors · 2023-11-19
> **Response to Reviewer YfM6**
>
> **General Response**:
> We sincerely appreciate the time you dedicated to reviewing our paper, and we are grateful for the valuable insights you provided.
> Before we get into the answers to detailed questions, we would like to emphasize the following three points.
>
> 1. Bilevel programs are challenging to solve both theoretically and numerically.
> In particular, for bilevel programs with binary tender as considered in this paper, the computational burden increases exponentially with respect to the number of linking variables.
> Therefore, for large instances, it is intractable to obtain the optimal solution.
> In this paper, we use the state-of-the-art solver, MiBS, as the benchmark to assess the solution quality of our method. Due to the intractability, even MiBS may fail to provide optimal solutions to some bilevel test instances.
> For this reason, we have replaced the term ''relative error'' with ''objective difference'' to avoid confusion.
>
> 2. In our method, sampling is required to obtain sufficient $(\hat{x}$, $\phi(\hat{x}))$ pairs for training.
> The proposed enhanced sampling consists of two steps, to first find a feasible $\hat{x}$ by solving (5) and then to compute $\phi(\hat{x})$ by solving the lower-level problem.
> We note that because we only need a feasible $\hat{x}$ in each iteration before attaining the final optimum, it is unnecessary to solve (5) to optimality, which significantly reduces the computational burden during sampling.
> The following sentence is added in Section 3.1 to state this point.
>
>     *''Here we note that though (5) is a mixed-integer quadratic program, we only need a feasible solution to (5) for sampling, which does not incur too much computational burden and saves the overall computational time.''*
>
> 3. This paper focuses on bilevel programs.
> Once we have reformulated bilevel programs into single-level ones, such as (12), we can utilize existing algorithms and highly effective commercial solvers for solution.
> For this reason, how to solve (12) to optimality is out of the scope of this paper.
> The following sentence is added in Section 3.1 to state this point.
>
>     *''Here we note that existing algorithms and highly-efficient off-the-shelf solvers can be adopted to handle the single-level program (12), which is yet out of the scope of this paper.''*
>
> With the above three points in mind, we respond to your comments and questions one by one in the following.
>
> > **Weakness 1**:
> The numerical experiments lack sufficient evidence to demonstrate the advantages of the proposed algorithm.
> It appears that the algorithm can only achieve optimality for test instances with $n=10$.
> The authors solely conduct comparisons with the baseline method MiBS based on computational times, without assessing solution quality, such as the optimality gap.
>
> **Response**:
> Thanks for your comments.
> We agree that we do not calculate the optimality gap in our numerical experiments, yet as we have emphasized in the general response, it is currently impossible to obtain the optimal solution to large-size bilevel programs.
> In this paper, we use the objective difference to assess the solution quality, and the results are reported in Section 4.
>
> > **Weakness 2**:
> The test instances are only from the authors' synthetic generated ones.
> Some public datasets like MIBLP-XU and IBLP-FIS in Tahernejad et al., 2020 should be used.
>
> **Response**:
> Thanks for your suggestion.
> Actually, the instance generation rule provided in Appendix C of this paper follows closely the rules of generating MIBLP-XU in Tahernejad et al., 2020.
> Because we focus on bilevel programs with binary tender, the existing instances in MIBLP-XU are not applicable.
> So we have used the instance generation rule of MIBLP-XU as the basis to generate new instances with binary tender.
> The following sentence is added in Appendix C to state this point.
>
> *''Following the instance generation rules of (Tahernejad et al., 2020), all instances are generated in the following forms.''*
>
>
> > **Weakness 3**:
> This method should only be suitable when functions $f$ and $g$ are linear or MILP-representable because one needs to solve Eq (5) and (12) to optimality.
>
> **Response**:
> Thanks for your comments.
> As we have emphasized in the general response, we do not need to solve (5) to optimality, and how to solve (12) to optimality is out of the scope of this paper, because (12) can be solved effectively by various off-the-shelf solvers.
> Therefore, we do not require $f$ or $g$ to be linear or MILP-representable when using our method.
>
> > **Weakness 4**:
> The cost to solve a single instance of Problem (5) is very expensive because it is at least as hard as a mixed-integer quadratic program (MIQP).
>
> **Response**:
> Thanks for your comments.
> As we have emphasized in our general response (ii) before, we do not need to solve (5) to optimality, which significantly reduces the computational burden during sampling.

---

> ### Author Response · Authors · 2023-11-19
> **Response to Reviewer YfM6 (continued)**
>
> > **Question 1**:
> What is the stopping criterion for Algorithm 2 so that the generated point $(x,y)$ is guaranteed to be feasible to Problem 1?
> That is, $x$, $y$ is satisfied $x\in X(y)$, $y\in Y(x)$, $g(y,x)\geq \max_{z\in Y(x)}g(z,x)$
>
> **Response**:
> Due to the assumption that the leader's feasible region is independent of $y^{\*}$, i.e., $X(y^{\*})\equiv X$, any generated point $(x,y)$ from Algorithm 2 is feasible, for which there is no extra stopping criterion.
> The following gives a detailed explanation.
>
> According to Step 8 of Algorithm 2, the obtained $x^{\*}$ must satisfy the constraints of (12), which means $x^{\*}\in X$.
> According to Step 9 of Algorithm 2, we fix $x=x^{\*}$ and solve the lower-level problem to obtain $y^{\*}$, which means $y^{\*}\in Y(x^{\*})$ and $g(y^{\*},x^{\*})=\max_{z\in Y(x^{\*})}g(z,x^{\*})$.
> Therefore, any generated point $(x,y)$ from Algorithm 2 must satisfy $x\in X$, $y\in Y(x)$, and $g(y,x)\geq\max_{z\in Y(x)}g(z,x)$.
> Under the assumption, the generated point $(x,y)$ is naturally feasible to Problem (1).
> The following sentence is highlighted to state the point.
>
> *''In addition, we assume the leader's feasible region is independent of $y^{\*}$, i.e., $X(y^{\*})\equiv X$, which is a special case of (1). In this case, we can obtain an upper bound for the BP (1) from any feasible solution $x$ and the corresponding follower's optimal solution $y^{\*}$ (e.g., by solving the follower's problem using $x$).''*
>
>
> > **Question 2**:
> What is the definition of ''relative error'' in Figures 4 and 7?
> How do you define an error when an optimal solution cannot be found, for example, for large-scale problems with $n = 60$?
>
> **Response**:
> Thanks for your question.
> As we have emphasized in our general response (i) before, we have replaced the term ''relative error'' with ''objective difference'' to avoid confusion.
> The objective difference is assessed by using the solution of MiBS as the benchmark.
> We also provide the definition of ''objective difference'' in the revised paper as
>
> *''We compare our method with the state-of-the-art solver for bilevel problems, MiBS (Tahernejad et al., 2020) and use its solution under a time limit of 1 hour as the benchmark to calculate the objective differences (i.e., the gap between the objectives from our method and MiBS).''*
>
> > **Question 3**:
> Why relative errors can be negative (as in Fig. 7)?
>
> **Response**:
> Thanks for your question. According to the definition provided in the response to Question 2, the objective difference (previously relative error) would be negative if our method provides a better solution than the benchmark.
> The following sentence is added in the revised paper for explanation.
>
> *''The negative objective difference means that our method provides a better solution than the benchmark.''*
>
> > **Question 4**:
> Why we need to care of supermodularity of an ISNN?
> I notice that ISNN requires more neurons compared to GNN (i.e., the lower bound value for $N_{nr}$ in Eq. (8) vs Eq. (7))
>
> **Response**:
> Thanks for your question.
> First, the number of neurons in GNN and ISNN depend on the values of $N_{s}$ and $n$ and so there is no guarantee that one needs more neurons than the other.
> Also, according to Proposition 3, many weight parameters of ISNN end up being zero after training. Hence, it may not be completely fair to compare the complexity of GNN and ISNN only through the number of neurons.
>
> Second, the supermodularity of an ISNN helps to improve the solution quality of our method.
> For bilevel programs with binary tender, many lower-level problems generate a supermodular $\phi(x)$ [R3, R4]. If we relax the binary limit $x\in${$\{0,1\}$} of $\hat{\phi}(x)$ to $x\in[0,1]$, $\hat{\phi}(x)$ becomes a lower envelope of the discrete $\phi(x)$.
> Consequently, if we adopt an ISNN, the trained $\tilde{\phi}(x)$ tends to be a lower bound of $\phi(x)$ (see Figure 3) so that with high probability, the optimal solution to the original problem would not be cut off by (12b).
> Conversely, if we adopt a GNN, there is no guarantee that $\tilde{\phi}(x)\geq\phi(x)$ or vice versa, for which the optimal solution to the original problem may be cut off by (12b).
> Therefore, the supermodularity of an ISNN helps to improve the solution quality of our method.
>
> [R3] X. Chen, D.Z. Long, J. Qi. Preservation of supermodularity in parametric optimization: necessary and sufficient conditions on constraint structures. Operations Research. 2021, 69(1): 1-12.
> [R4] D.Z. Long, J. Qi, A. Zhang. Supermodularity in two-stage distributionally robust optimization. Management Science, ahead of print.
>
> > **Question 5**:
> Do we have to solve $\max_{y\in Y(y,\hat{x})}$ to optimality in Algorithm 1?
>
> **Response**:
> Thanks for your question.
> Yes, in step 7 of Algorithm 1, we need to solve $\max_{y\in Y(y,\hat{x})}$ to optimality to obtain correct $(\hat{x}$, $\phi(\hat{x}))$ pairs for training.

---

> ### Author Response · Authors · 2023-11-19
> **Response to Reviewer YfM6 (continued)**
>
> > **Question 6**:
> For feasibility purpose, why do we need the second term $h^{\textrm{T}}x$ in Eq. (5)?
>
> **Response**:
> Thanks for your question.
> The objective (5a) is a general formula for quadratic programming, which does not affect feasibility.
> In other words, we can set $h = 0$ in (5) without affecting the feasibility.
>
> > **Question 7**:
> I don't fully understand the notation ''GNN representability'' in Prop 1 (ii).
> This statement ''we measure the GNN representability by the number of parameters it trains'' seems to be not mathematically sound.
>
> **Response**:
> Thanks for your comments.
> In the studies about the representability of neural networks, the number of trainable parameters is a very straightforward metric, which is accepted but is also criticized by [R5] due to its conservativeness when $x$ lies in a continuous domain.
> Instead, [R5] recommends using the number of linear regions of the graph $(x, \tilde{\phi}(x))$.
> However, as we study bilevel programs with binary tender in this paper, the domain of $x$ is discrete (in particular, {$\{0,1\}$}$^n$).
> In this case, the graph $(x, \tilde{\phi}(x))$ becomes a collection of points and it is no longer well-defined to measure its representability using the number of linear regions.
> Instead, the number of trainable parameters is more applicable and is thus used to measure the representability of neural networks in this paper.
>
> [R5] Hu, X., Chu, L., Pei, J. et al. Model complexity of deep learning: a survey. Knowl Inf Syst 63, 2585--2619 (2021).
>
> > **Question 8**:
> The architecture details of GNN of ISNN should be reported.
>
> **Response**:
> Thanks for your comments.
> In Section 3.2, we have reported the architecture of GNN of ISNN (see below also).
>
> ''According to Proposition 1, when we adopt a GNN for training, we incorporate two hidden layers in the GNN architecture, each with $N_{nr}/2$ neurons, where $N_{nr}$ is the smallest even number that satisfies (7).''
>
> ''Since the ISNN’s architecture does not affect its representability, to better compare with GNN, we incorporate two hidden layers in ISNN, each with $N_{nr}/2$ neurons, where $N_{nr}$ is the smallest integer that satisfies (8).''
>
> > **Question 9**:
> The name of section 3.3 seems to be incomplete.
>
> **Response**:
> Thanks for your comments.
> The main steps of our method consist of sampling, training, and solving, which are used as the titles of sections 3.1-3.3, respectively.
>
> > **Question 10**:
> I don't see any part of Figures to support the claims ''achieving relative error less than 5%,'', ''the relative error of instances with n = 30, 50, 60 is larger than 15%.''
>
> **Response**:
> Thanks for your comments.
> For the claim ''the relative error of instances with n = 30, 50, 60 is larger than 15\%,'' please see Figure 4.
> For the claim ''achieving relative error less than 5\%,'' please see Figure 7.

---

> ### Author Response · Authors · 2023-11-22
> **looking forward to post-rebuttal feedback!**
>
> Dear Reviewer YfM6
>
> Thank you for reviewing our paper. We have carefully answered your concerns about optimality, datasets, and all questions.
>
> Please let us know if our answers accurately address your concerns. If our response resolves your concerns, we kindly ask you to consider raising the rating of our work. Thank you very much for your time and efforts! We would like to discuss any additional questions you may have.
>
> Best, Authors

---

### Official Review · Reviewer_nqbY · 2023-10-31

**Soundness:** 2 fair
**Presentation:** 2 fair
**Contribution:** 2 fair
**Rating:** 6
**Confidence:** 3

**Summary:**

This work studies the Bilevel programs (BPs)  with discrete decision variables. A neural network is trained to approximate the optimal value of the lower-level problem, as a function of the binary tender. Then a single-level reformulation of the BP through a mixed-integer representation of the value function can be obtained. Moreover, an enhanced sampling method is proposed for high-dimensional BPs.

**Strengths:**

1. This work proposes an approximation-based method for Bilevel programs (BPs)  with discrete decision variables, which is interesting.

2. an input supermodular neural network (ISNN) is proposed, which ensures a supermodular mapping from input to output.

3. an enhanced sampling method is proposed for solving high-dimensional BPs.

**Weaknesses:**

1. The author should conduct some complexity analysis, such as time complexity [1], to show the effectiveness of the proposed method.

2. This work employs neural networks to learn and approximate the value function $\phi(x)$. However, training the neural networks is more computationally complex than directly approximating the lower-level optimization problem [2]. What is the advantage of the proposed method over the polyhedral approximation in [2]?

3. Since one of the key contributions in this work is to employ neural networks to learn and approximate the value function $\phi(x)$, I suggest the authors clearly discuss the existing approximation-based methods (i.e., which approximate the lower-level optimization problems and reformulate the bilevel optimization problems as single-level optimization problems, for instance, polyhedral approximation) in bilevel optimization since I can't find any discussion about the approximation-based methods in bilevel optimization.

4. Training the neural networks to approximate the value function $\phi(x)$ may introduce some variance which will lead to the solution of the resulting single-level problem far away from the original bilevel optimization problems. Can you provide a more theoretical guarantee for the proposed method?

[1] A Gradient Method for Multilevel Optimization. NeurIPS 2021.

[2] Asynchronous Distributed Bilevel Optimization, ICLR 2023.

**Questions:**

I have some questions about the complexity, the comparison with the existing approximation-based methods in bilevel optimization, and the theoretical guarantee of the proposed method. Please see the Weakness.

---

> ### Author Response · Authors · 2023-11-21
> **Response to Reviewer nqbY**
>
> **General Response**:
> We sincerely appreciate the time you dedicated to reviewing our paper, and we are grateful for the valuable insights you provided.
> Before answering your questions, we would like to emphasize the following two points.
>
> 1. This paper focuses on bilevel programs with binary tender.
> It means that the input $x$ of the value function $\phi(x)$ is binary-valued, to which existing gradient methods are no longer applicable.
>
> 2. Although gradient methods are not applicable, there exist works that seek to approximate $\phi(x)$ using cutting planes.
> To our best knowledge, MiBS [R1] is the state-of-the-art among these works and is thus adopted as the benchmark in the numerical experiments of this paper.
>
>     [R1] Sahar Tahernejad, Ted K Ralphs, and Scott T DeNegre. A branch-and-cut algorithm for mixed integer bilevel linear optimization problems and its implementation. Mathematical Programming Computation, 12(4): 529--568, 2020.
>
> With the above two points in mind, we respond to your comments and questions one by one in the following.
>
>
> > **Weakness 1**:
> The author should conduct some complexity analysis, such as time complexity [1], to show the effectiveness of the proposed method.
> [1] A Gradient Method for Multilevel Optimization. NeurIPS 2021.
>
> **Response**:
> Thanks for your suggestion.
> We have carefully read the recommended paper [1], in which a gradient method is proposed to solve multilevel optimization and the time and space complexities of the gradient method are analyzed.
> However, as we have emphasized in the general response, gradient methods as well as the time and space complexity analyses are not applicable to the class of bilevel integer programs in this paper, for which the time complexity analysis for gradient methods in [1] is inapplicable.
>
>
> > **Weakness 2**:
> This work employs neural networks to learn and approximate the value function $\phi(x)$.
> However, training the neural networks is more computationally complex than directly approximating the lower-level optimization problem [2].
> What is the advantage of the proposed method over the polyhedral approximation in [2]?
> [2] Asynchronous Distributed Bilevel Optimization, ICLR 2023.
>
> **Response**:
> Thanks for your comments.
> First, we have carefully read the recommended paper [2], in which cutting planes based on [R2] are utilized to form polyhedral approximations.
> Specifically, [2] construct these cutting planes using gradient information from the lower-level problem.
> However, as we have emphasized in the general response, the class of bilevel programs we study in this paper have binary tenders and the lower-level problem therein contain binary decision variables.
> For these reasons, gradient methods become inapplicable, and so do the polyhedral approximation in [2].
>
> Second, since the lower-level problem in this paper involves binary decision variables, it is in general inapproximable (see, e.gl, [R3]), that is, there do not exist polynomial-time algorithms that produce solutions to such lower-level problems with a constant competitive ratio.
>
> Third, we clarify that this paper does not intend to prove that neural networks outperform any approximation methods in solving general bilevel programs.
> Instead, in view of the inapplicability of the existing approximation methods (see above) and the power of neural networks in other machine learning problems, we seek to demonstrate that NN also help solve this challenging class of bilevel programs.
>
> [R2] Stephen Boyd and Lieven Vandenberghe. Localization and cutting-plane methods. From Stanford EE 364b lecture notes, 2007.
> [R3] Trevisan, Luca. Inapproximability of combinatorial optimization problems. Paradigms of Combinatorial Optimization: Problems and New Approaches (2014): 381-434.

---

> > ### Author Response · Authors · 2023-11-21
> > **Response to Reviewer nqbY (continued)**
> >
> > > **Weakness 3**:
> > Since one of the key contributions in this work is to employ neural networks to learn and approximate the value function $\phi(x)$, I suggest the authors clearly discuss the existing approximation-based methods (i.e., which approximate the lower-level optimization problems and reformulate the bilevel optimization problems as single-level optimization problems, for instance, polyhedral approximation) in bilevel optimization since I can't find any discussion about the approximation-based methods in bilevel optimization.
> >
> > **Response**:
> > Thanks for your suggestion.
> > Actually, we have discussed some approximation-based methods in the part ''Bilevel (mixed-integer) nonconvex programs'' of Section 2.
> > In this revision, as suggested, we have modified related sentences to highlight such methods.
> >
> > *''For BPs with nonconvex objectives/constraints, approximation methods were employed to approximate the lower-level problem and reformulate the BP as a single-level formulation, such as polyhedral approximation (Sato et al., 2021) and other gradient-based methods in machine learning (Hong et al., 2020; Chen et al., 2022).
> > However, when discrete decision variables are incorporated, gradient-based methods are no longer applicable.
> > To address this issue, integer programming techniques are employed and various cutting plane-based algorithms, such as the MiBS solver, have been developed (Ralphs et al., 2015; Zeng \& An, 2014).''*
> >
> > > **Weakness 4**:
> > Training the neural networks to approximate the value function $\phi(x)$ may introduce some variance which will lead to the solution of the resulting single-level problem far away from the original bilevel optimization problems.
> > Can you provide a more theoretical guarantee for the proposed method?
> >
> > **Response**:
> > Thanks for your question.
> > We agree with you that an inaccurate approximation $\tilde{\phi}$ may lead to low-quality solutions to the bilevel program.
> > In response, we have provided a theoretical guarantee for the approximation error of $\tilde{\phi}$.
> > Please see the details in Appendix A.4.
> >
> > > **Questions**:
> > I have some questions about the complexity, the comparison with the existing approximation-based methods in bilevel optimization, and the theoretical guarantee of the proposed method.
> > Please see the Weakness.
> >
> > **Response**:
> > Thanks for your questions.
> > In the responses to the Weakness, we have carefully answered the related questions about the complexity, the comparison with the existing approximation-based methods, and the theoretical guarantee.
> > Please see the above responses for details.

---

> > > ### Comment · Reviewer_nqbY · 2023-11-22
> > >
> > > Thanks for responses, my concerns have been addressed and I'm happy to change the score to 6.

---

> > > > ### Author Response · Authors · 2023-11-22
> > > >
> > > > We sincerely thank the reviewer for your valuable comments and appreciate the increase in scores.

---

### Official Review · Reviewer_skLM · 2023-10-31

**Soundness:** 2 fair
**Presentation:** 3 good
**Contribution:** 2 fair
**Rating:** 6
**Confidence:** 4

**Summary:**

This paper employs neural networks to help solve the bilevel programs with binary tenders. Specifically, they adopt neural networks to approximate the optimal value of the lower-level problem as a function of the binary tenders. In order to train the neural networks, the authors also introduce an enhanced sampling method to generate high-quality samples. Lastly, numerical experiments were conducted to demonstrate the performance of these approaches.

**Strengths:**

This paper gives a good attempt of incorporating ML (especially neural networks) methods to facilitate the solution of traditional mathematical optimization problems, which in my opinion is an area that deserves more attention from the community. Overall, this paper is clearly written, easy to understand, and the theoretical results in Section 3 are very neat. I also find it to be very impressive that I cannot find any typo throughout this paper and the propositions also appear to be of their own independent interest.

**Weaknesses:**

Main concern:
1. The lack of ablation study, especially on the enhanced sampling part. For instance, why do you want to solve the quadratic programming problem (5) to get the samples? I understand that matrix Q is selected to be PSD is for the polynomial-solvability, but what is the main reason of solving the quadratic program in the first place? If we replaced this enhanced sampling with some other more naive sampling methods, how would it affect the experiment results?
2. Limitation of the experiment setup and analysis:
   (a) Instance dimension up to 60 is too small.
   (b) The selection of the lower level problem is LP and MILP, which both have linear lower level objective function. At least you could have tried some simple nonlinear functions like quadratic function.
   (c) The experimental results do not support the conclusion well. I list some of the points in the next Questions section.

**Questions:**

Major questions:
1. In Figure 5 and Figure 6, the computational time of MiBS increases very fast with the increase of n. My question is, even though the MiBS solver might take a long time to reach optimality, have you considered to set a time limit, and compare the relative error of the best found solution given by the solver at time limit with your approaches?
2. In your Conclusion section you mentioned: "we demonstrated that the enhanced sampling helps reduce average relative error". However, the whole point of your sampling method is simply to get enough data points for training the neural network. In order to claim that your proposed enhanced sampling can bring some extra benefits, you need to at least compare with other non-trivial sampling methods.
3. In Conclusion section: "The computational time of using ...... is significantly shorter than that of MiBS ......" I admit that this is true, but your methods also do not reach optimality, for fair comparison you either need to have enough samples for exactly learning the value function (so that your approach can also produce a true optimal solution), or you need to compare the best feasible solution found by MiBS within a given time limit.

Minor question:
1. I suggest to give an exact definition for "binary tender". As far as I know, I do not think this is a well-known term in the community.
2. Still about the definition of "binary tender". On Page 2 "we assume that the entries of x appearing in the lower-level formulation are binary-valued". However, in Appendix C, about the instance generation for the experiments, I notice that the binary constraint on x is enforced in the upper-level instead of the lower-level formulation.

---

> ### Author Response · Authors · 2023-11-20
> **Response to Reviewer skLM**
>
> **General Response**:
> We sincerely appreciate the time you dedicated to reviewing our paper, and we are grateful for the valuable insights you provided.
> Before responding to your questions, we would like to emphasize the following two points.
>
> 1. Sampling sufficiently many $(\hat{x}, \phi(\hat{x}))$ pairs is fundamental to our approach, which replies on the approximation of the value function $\phi(x)$.
> However, as $n$ gets larger, due to the exponentially increasing input space, feasible $x$ solutions become more scarce and it gets harder to obtain samples with respect to feasible, let alone near-optimal, $x$ solutions.
> For this reason, naive sampling approaches quickly becomes inapplicable as $n$ increases.
> Instead, through the ''sampling via optimization'' approach developed in this paper, we guarantee finding a feasible $x$ in each sampling.
> By solving a quadratic program (not necessarily to global optimium), we can avoid repeated samples and enhance sampling efficiency.
> Furthermore, by imposing constraints like (5b), we can enhance the quality of the obtained samples which improves the quality of the approximate value function $\tilde{\phi}(x)$.
>
> 2. In all numerical experiments, we set the time limit of MiBS to be 1 hour in this paper.
>
> With the above two points in mind, we respond to your comments and questions one by one in the following.
>
> > **Weakness 1**:
> The lack of ablation study, especially on the enhanced sampling part.
> For instance, why do you want to solve the quadratic programming problem (5) to get the samples?
> I understand that matrix $Q$ is selected to be PSD is for the polynomial-solvability, but what is the main reason of solving the quadratic program in the first place?
> If we replaced this enhanced sampling with some other more naive sampling methods, how would it affect the experiment results?
>
> **Response**:
> Thanks for your comments.
> We answer the questions one by one as follows.
>
> 1. > If we replaced this enhanced sampling with some other more naive sampling methods, how would it affect the experiment results?
>
>     If we replace enhanced sampling with naive sampling, as we have emphasized in the general response point 1, we cannot obtain sufficient feasible $x$ within a reasonable time limit, when the dimension of $x$ gets large.
>     As a result, the quality of the approximate value function $\tilde{\phi}(x)$ will be poor and producing low-quality solutions to the bilevel program.
>
> 2. > why do you want to solve the quadratic programming problem (5) to get the samples?
>
>     The reason for solving a quadratic program (5), as opposed to for example a linear program, is to avoid repeated samples and enhance sampling efficiency.
>     If we solve a linear program to get samples, i.e., removing the term $x^{\textrm{T}}Qx$ in (5a), all $x$ thus obtained are guaranteed to lie on the boundary of the feasible region (or the convex hull of the feasible region).
>     As a result, these $x$-solutions become increasingly repetitive as the dimension $n$ increases, decreasing the sampling efficiency.
>     Instead, when (5) has a quadratic (and convex) objective function, the optimal solution may lie both on the boundary or in the interior of the feasible region (or the convex hull of the feasible region).
>     This drastically enhances the sampling efficiency.
>     The following sentence is added in Section 3.1 to state this point.
>
>     *''We note that the reason for solving a quadratic program here is to avoid repeated samples and enhance sampling efficiency.''*
>
> 3. > What is the main reason of solving the quadratic program in the first place?
>
>     There are two main reasons here.
>     The first reason is to enhance sampling efficiency, as discussed above.
>     The second reason is that we only need a feasible solution to (5), instead of an optimal solution.
>     Since most commercial solvers employ advanced heuristics to find feasible solutions to mixed-integer quadratic programs, formulating (5) as a MIQP, as opposed to other mixed-integer convex programs, further enhances the sampling efficiency.
>     The following sentence is added in Section 3.1 to state this point.
>
>     *''Here we note that though (5) is a mixed-integer quadratic program, we only need a feasible solution to (5) for sampling, which does not incur too much computational burden and saves the overall computational time.''*

---

> > ### Author Response · Authors · 2023-11-20
> > **Response to Reviewer skLM (continued)**
> >
> > > **Weakness 2**:
> > Limitation of the experiment setup and analysis:
> > (a) Instance dimension up to 60 is too small.
> > (b) The selection of the lower level problem is LP and MILP, which both have linear lower level objective function.
> > At least you could have tried some simple nonlinear functions like quadratic function.
> > (c) The experimental results do not support the conclusion well.
> > I list some of the points in the next Questions section.
> >
> > **Response**:
> > Thanks for your comments.
> > We address the mentioned limitations one by one.
> >
> > 1. > Instance dimension up to 60 is too small.
> >
> >     We would like to emphasize that bilevel programs are challenging to solve, and especially for bilevel programs with binary tender (which is considered in this paper), the computational burden increases exponentially with respect to the number $n$ of the linking variables.
> >     Therefore, $n=60$ is not a small instance for the class of problems we are studying, which are challenging bilevel binary-integer programs. .
> >     Nevertheless, we have added new instances with $n$ up to 120 in the revised paper.
> >     Please see Appendix D.4 for details.
> >
> > 2. > The selection of the lower level problem is LP and MILP, which both have linear lower level objective function. At least you could have tried some simple nonlinear functions like quadratic function.
> >
> >     We remark that our approach applies to lower-level problems with an arbitrary objective function.
> >     In fact, we agree with you and look forward to testing our approach in, e.g., quadratic instances.
> >     However, as the state-of-the-art and the benchmark method to compare to, MiBS is applicable to linear lower-level problems only.
> >     Consequently, we have to restrict ourselves to linear lower-level problems at this point and look forward to testing our approach in more general instances when a proper benchmark solver becomes available.
> >
> > 3. > The experimental results do not support the conclusion well. I list some of the points in the next Questions section.
> >
> >     We have carefully addressed all the listed points below.
> >     Please see corresponding answers for details.
> >
> >
> > > **Major question 1**:
> > In Figure 5 and Figure 6, the computational time of MiBS increases very fast with the increase of n.
> > My question is, even though the MiBS solver might take a long time to reach optimality, have you considered to set a time limit, and compare the relative error of the best found solution given by the solver at time limit with your approaches?
> >
> > **Response**:
> > Thanks for your question.
> > Actually, we have set the time limit of MiBS to be 1 hour and the relative error in all existing results is compared to the best-found solution of MiBS within this time limit.
> > The following sentence is added in Section 4.2.2 to state this point.
> >
> > *''We compare our method with the state-of-the-art solver for bilevel problems, MiBS (Tahernejad et al., 2020) and use its solution within a 1-hour time limit as the benchmark to calculate the objective differences (i.e., the gap between the objectives from our method and MiBS).''*
> >
> > In this revision, we also vary the length of the time limit of MiBS to 0.5 hours, 1.5 hours, and 2 hours, respectively.
> > But for each instance used in the paper, the best-found solution of MiBS keeps the same under different time limits.

---

> ### Author Response · Authors · 2023-11-20
> **Response to Reviewer skLM (continued)**
>
> > **Major questions 2**:
> In your Conclusion section you mentioned: ``we demonstrated that the enhanced sampling helps reduce average relative error".
> However, the whole point of your sampling method is simply to get enough data points for training the neural network.
> In order to claim that your proposed enhanced sampling can bring some extra benefits, you need to at least compare with other non-trivial sampling methods.
>
> **Response**:
> Thanks for your comments.
> The proposed enhanced sampling brings us two aspects of benefits: sampling efficiency and sample quality.
>
> On the one hand, as we have emphasized in the general response, as compared to naive sampling which draws samples of $x$ from {$\{0,1\}$}$^n$ based on a certain probability distribution, the adopted ''sampling via optimization'' approach guarantees finding a feasible $x$ in each sampling, which enhances the sampling efficiency.
> In this revision, we compare the sampling efficiency of our method with a naive sampling method (random sampling) and a non-trivial sampling method (Latin hypercube sampling) and provide the results in Appendix D.3.
> It is obvious that our sampling method is more stable and applicable than both benchmarks.
>
> On the other hand, by adding and updating the constraint (5b), we can sample more often in the vicinity of the optimal solution, which enhances the quality of the found samples.
> Actually, we have validated this point in our numerical experiments (see Figure 4 and Figure 6).
> According to Algorithm 2, $N_{iteration}=1$ means no constraint (5b), $N_{iteration}=2$ means adding a constraint (5b), and $N_{iteration}=3$ means updating the constraint (5b).
> We observe that a larger $N_{iteration}$ helps to reduce the objective difference, which corresponds to the mentioned sentence ''we demonstrated that the enhanced sampling helps reduce average relative error".
>
> > **Major questions 3**:
> In Conclusion section: ''The computational time of using ...... is significantly shorter than that of MiBS ......"
> I admit that this is true, but your methods also do not reach optimality, for fair comparison you either need to have enough samples for exactly learning the value function (so that your approach can also produce a true optimal solution), or you need to compare the best feasible solution found by MiBS within a given time limit.
>
> **Response**:
> Thanks for your comments.
> We agree with the comment and in this paper, we have chosen the latter suggestion, which compares the best feasible solution found by MiBS within a given time limit.
> Please see our response to Major Question 1 for more details.
>
>
> > **Minor questions 1**:
> I suggest to give an exact definition for ''binary tender".
> As far as I know, I do not think this is a well-known term in the community.
>
> **Response**:
> Thanks for your suggestion.
> In the revised paper, a more exact definition for ''binary tender" is provided as
>
> *''Here, ''tender'' is defined as the linking variables between the upper- and lower-level problems and ''binary tender'' means that all tender variables are binary.''*
>
> > **Minor questions 2**:
> Still about the definition of ''binary tender".
> On Page 2 "we assume that the entries of x appearing in the lower-level formulation are binary-valued".
> However, in Appendix C, about the instance generation for the experiments, I notice that the binary constraint on x is enforced in the upper-level instead of the lower-level formulation.
>
> **Response**:
> Thanks for your comments.
> We would like to remark that the lower-level problem can be seen as a parametric optimization problem, where the upper-level variables $x$ are treated as given parameters.
> Therefore, the binary constraint on $x$ should be enforced in the upper level, but not in the lower level where $x$ becomes parameters.

---

> ### Author Response · Authors · 2023-11-22
> **looking forward to post-rebuttal feedback!**
>
> Dear Reviewer skLM
>
> Thank you for reviewing our paper. We have carefully answered your concerns about sampling, time limits, and experimental setup.
>
> Please let us know if our answers accurately address your concerns. If our response resolves your concerns, we kindly ask you to consider raising the rating of our work. Thank you very much for your time and efforts! We would like to discuss any additional questions you may have.
>
> Best, Authors

---

> ### Comment · Reviewer_skLM · 2023-11-23
>
> Regarding my "Weakness 1", I'm still not fully convinced by your explanation. I cannot see the necessity of solving a separating quadratic programming simply to get some feasible point in the interior of the feasible region. You mentioned "as n gets larger, due to the exponentially increasing input space, feasible solutions become more scarce and it gets harder to obtain samples with respect to feasible", but since the feasible region does not only have finitely many element, you can totally design some much easier sampling method based on the formulation of the feasible region. That being said, I appreciate your detailed response to my questions and it has addressed most of my other concerns. Therefore, I decide to raise my point to 6.

---

> > ### Author Response · Authors · 2023-11-23
> >
> > Thanks for your kind comment!
> > There may exist various approaches to get feasible points in the interior of the feasible region.
> > Adding a second-order term in the objective is a straightforward approach and is adopted in the paper.
> > But we also agree that modifying the formulation of the feasible region, as you suggested, is another approach.
> > Actually, we have implemented this idea in our sampling process, i.e., adding constraint (5b) to narrow the feasible region (though (5b) is designed in this paper to enhance sample quality rather than sampling efficiency).
> > We thank the reviewer's suggestion and will consider the latter approach in our following works.
> >
> > Lastly, we sincerely thank the reviewer for your valuable comments and appreciate the increase in scores.

---

### Meta-Review · Area_Chair_8dou · 2023-12-03

**Metareview:**

This paper studies a challenging bilevel optimization problem, in which both upper-level and lower-level problems may involve discrete decision variables. Due to the presence of discrete variables, existing bilevel optimization algorithms based on hypergradient cannot be applied. The authors propose to learn the value function of the lower-level problem using a neural network. The entire bilevel problem is then reduced to a mixed integer program, which can be solved using off-the-shelf MIP solvers. The numerical results are impressive. Overall, this is a solid work with interesting ideas and results.

**Justification For Why Not Higher Score:**

Not significant enough for a higher score.

**Justification For Why Not Lower Score:**

Good results. Should be accepted.

---

### Decision · Program_Chairs · 2024-01-16

Accept (poster)